# Desensitized chimeric antigen receptor T cells selectively recognize target cells with enhanced antigen expression

Chungyong Han [1], Su-Jung Sim[1], Seon-Hee Kim[1], Rohit Singh[1], Sunhee Hwang[2], Yu I. Kim[3], Sang H. Park[1], Kwang H. Kim[1], Don G. Lee[4], Ho S. Oh[2], Sangeun Lee[1], Young H. Kim[2,4], Beom K. Choi[4] & Byoung S. Kwon[2,5]

Chimeric antigen receptor (CAR) T cell therapy is an effective method for treating specific cancers. CARs are normally designed to recognize antigens, which are highly expressed on malignant cells but not on T cells. However, when T cells are engineered with CARs that recognize antigens expressed on the T cell surface, CAR T cells exhibit effector function on other T cells, which results in fratricide, or killing of neighboring T cells. Here, using human leukocyte antigen-DR (HLA-DR)-targeted CAR T cells, we show that weak affinity between CAR and HLA-DR reduces fratricide and induces sustained CAR downregulation, which consequently tunes the avidity of CAR T cells, leading to desensitization. We further demonstrate that desensitized CAR T cells selectively kill Epstein-Barr virus-transformed B cells with enhanced HLA-DR expression, while sparing normal B cells. Our study supports an avidity-tuning strategy that permits sensing of antigen levels by CAR T cells.

[1] Immunotherapeutics Branch, Division of Convergence Technology, Research Institute, National Cancer Center, Goyang 10408, Republic of Korea. [2] Eutilex Institute for Biomedical Research, Eutilex Co., Ltd., Seoul 08594, Republic of Korea. [3] Graduate School of Cancer Science and Policy, National Cancer Center, Goyang 10408, Republic of Korea. [4] Biomedicine Production Branch, Research Institute, National Cancer Center, Goyang 10408, Republic of Korea. [5] Department of Medicine, Tulane University Health Sciences Center, New Orleans, LA 70118, USA. Correspondence and requests for materials should be addressed to B.S.K. (email: bskwon@eutilex.com)

T cells engineered with chimeric antigen receptors (CAR T cells) have a great therapeutic potential for treating cancers[1–5]. Their clinical success is attributed to the fusion structure of the CAR, which is made by artificially combining a high-affinity antigen-binding domain with multiple signaling domains[6,7]. However, CARs frequently target antigens that are not exclusively expressed on malignant cells, but also expressed on normal cells (occasionally on T cells themselves). These differ from the T cell receptor (TCR), a natural antigen receptor for T cells, which typically shows low affinity and recognizes antigens rarely expressed on normal cells. Despite these differences, some properties of CARs are shared with TCRs.

One of the shared properties is receptor downregulation. TCRs are rapidly downregulated after antigen recognition to limit excess signaling to maintain signal integrity[8,9]. Similarly, antigen recognition by CARs is immediately followed by CAR down-regulation, which affects subsequent antigen recognition and function[10,11]. These events occur within hours and recover in days. In contrast to short-term downregulation, long-term downregulation was reported by Gallegos et al.[12]. The study demonstrated that continuous TCR–target interactions induced long-term TCR downregulation, which could be sustained for over 50 days. The extent of downregulation was correlated with TCR–target affinity and, most importantly, eventually resulted in an increase in the overall immune-activation threshold. This phenomenon represents a mechanism by which T cells tune antigen sensitivity and manage the extent of the immune response at the macro level. For CAR T cells, however, long-term CAR downregulation and subsequent functional changes induced by continuous target recognition have not been widely investigated.

While receptor downregulation is observed in both CARs and TCRs, the specific binding characteristics of CARs may result in a distinctive functional consequence known as "fratricide", which is T cell death induced by neighboring CAR T cells due to targeting of the antigen expressed on T cells. Interestingly, the extent of fratricide is not the same for all CAR constructs. Fratricide is transient in CD5-targeted CAR T cells[13], as they expand normally for several weeks. In contrast, fratricide seriously damages CD7-targeted CAR T cells, resulting in unviability[14]. However, the conditions that allow the extent of fratricide to be tolerable are not well-defined.

Here, we show that human leukocyte antigen-DR (HLA-DR)-targeted MVR CAR T cells continuously recognize HLA-DR on neighboring CAR T cells and induce fratricide and CAR down-regulation. Importantly, as MVR CAR recognizes the poly-morphic region of HLA-DR, T cells with different HLA-DRB1 alleles exhibit severe or mild degrees of fratricide and CAR downregulation depending on the strength of the binding affi-nities between HLA-DR and MVR CAR. We demonstrate that fratricide is reduced to a tolerable level when CAR–antigen affi-nity is low. Furthermore, we show that 'autotuning', a sensitivity tuning mechanism characterized by sustained CAR down-regulation, endows MVR CAR T cells with target-cell selectivity based on antigen level.

## Results

### Low CAR affinity reduces fratricide of MVR CAR T cells.
To investigate the effect of the interaction between CARs and T cell-derived antigens on fratricide and CAR downregulation, we used HLA-DR-targeted CAR T cells. HLA-DR, the clas-sical major histocompatibility complex II molecule, is expressed on antigen-presenting cells and activated T cells[15]. Because activated T cells express HLA-DR on their surface, T cells transduced with the CAR continuously recognize

HLA-DR and induce fratricide and CAR downregulation. The previously developed HLA-DR-specific antibody clone MVR was used to design an MVR CAR construct. Notably, as MVR recognizes the polymorphic region of HLA-DR, donors with different HLA-DRB1 alleles exhibited strong, intermediate, or weak binding with MVR (corresponding HLA-DRB1 alleles were designated as $DR^{str}$, $DR^{int}$, and $DR^{weak}$, respectively; Fig. 1a). Based on the distinctive binding behavior of MVR, we evaluated the extent of fratricide and CAR downregulation as a function of CAR–antigen affinity. We transduced $DR^{str}$, $DR^{int}$, and $DR^{weak}$ T cells with a second-generation MVR CAR construct (Fig. 1b). CD19-targeted CAR T (CD19 CAR T) cells and non-transduced T (NT T) cells were generated as controls. The growth rates and viability of $DR^{str}$ and $DR^{int}$ MVR CAR T cells were compromised, whereas $DR^{weak}$ MVR CAR T cells continued to grow in a similar manner to par-ental NT T cells (Fig. 1c, d). The frequency of MVR CAR-positive cells was profoundly decreased in $DR^{str}$ and $DR^{int}$ MVR CAR T cells, indicating that the interaction between MVR CAR and HLA-DR is crucial for fratricidal cell death (Fig. 1e).

Despite the limited fratricide of $DR^{weak}$ MVR CAR T cells, the interaction between $DR^{weak}$ HLA-DR and MVR CAR may result in continuous CAR signaling, giving rise to T cell exhaustion and related T cell dysfunction[16,17]. Therefore, the expression of representative exhaustion markers, LAG-3, TIM-3, CTLA-4, and PD-1[18-20], was examined in $DR^{int}$ and $DR^{weak}$ MVR CAR T cells. $DR^{weak}$ MVR CAR T cells did not display strong exhaustion or express multiple exhaustion markers simultaneously, in contrast to most $DR^{int}$ MVR CAR T cells (Fig. 1f and Supplementary Fig. 1). These data indicate that fratricide and exhaustion caused by the interaction of MVR CAR and HLA-DR are minimal in $DR^{weak}$ MVR CAR T cells, whereas they are severe in $DR^{str}$ and $DR^{int}$ MVR CAR T cells. All MVR CAR T cells used in the following sections were $DR^{weak}$ MVR CAR T cells, unless otherwise specified.

### CAR–HLA-DR interaction downregulates surface MVR CAR.
While $DR^{str}$ and $DR^{int}$ MVR CAR T cells exhibited heavy downregulation of CAR (Fig. 1e), $DR^{weak}$ MVR CAR T cells exhibited approximately 2-fold lower surface CAR expression than CD19 CAR T cells (Figs. 1e, 2a). This difference was con-firmed in 293T cell lines and primary $DR^{weak}$ T cells transduced with various multiplicities of infection of MVR CAR or CD19 CAR lentiviral vectors. Although surface MVR CAR expression increased with the multiplicity of infection in 293T cell lines, expression in primary $DR^{weak}$ T cells was unchanged (Fig. 2b). Longitudinal analysis of CAR expression revealed that $DR^{weak}$ T cells expressing the highest levels of surface MVR CAR were present 2 days post-transduction (4 days post-activation), and MVR CAR was gradually downregulated over the 14 days of the T cell activation cycle (Fig. 2c).

CAR mRNA and protein levels in $DR^{weak}$ MVR CAR T cells were similar to or higher than in CD19 CAR T cells, indicating that surface CAR is downregulated post-translationally (Fig. 2d, e and Supplementary Fig. 2). To determine if downregulation of MVR CAR was induced by the interaction of MVR CAR with HLA-DR, we attempted to generate HLA-DR-deficient MVR CAR T cells using the CRISPR-Cas9 system. However, we repeatedly failed, possibly because of an unknown survival advantage of HLA-DR in T cells. We therefore generated Epstein-Barr virus-induced lymphoblastoid cell lines defective in HLA-DR (ΔDR-EBV LCLs) and transduced these cells with MVR CAR lentivirus. ΔDR-EBV LCLs expressed higher levels of MVR CAR than $DR^{weak}$ EBV LCLs, and expression decreased

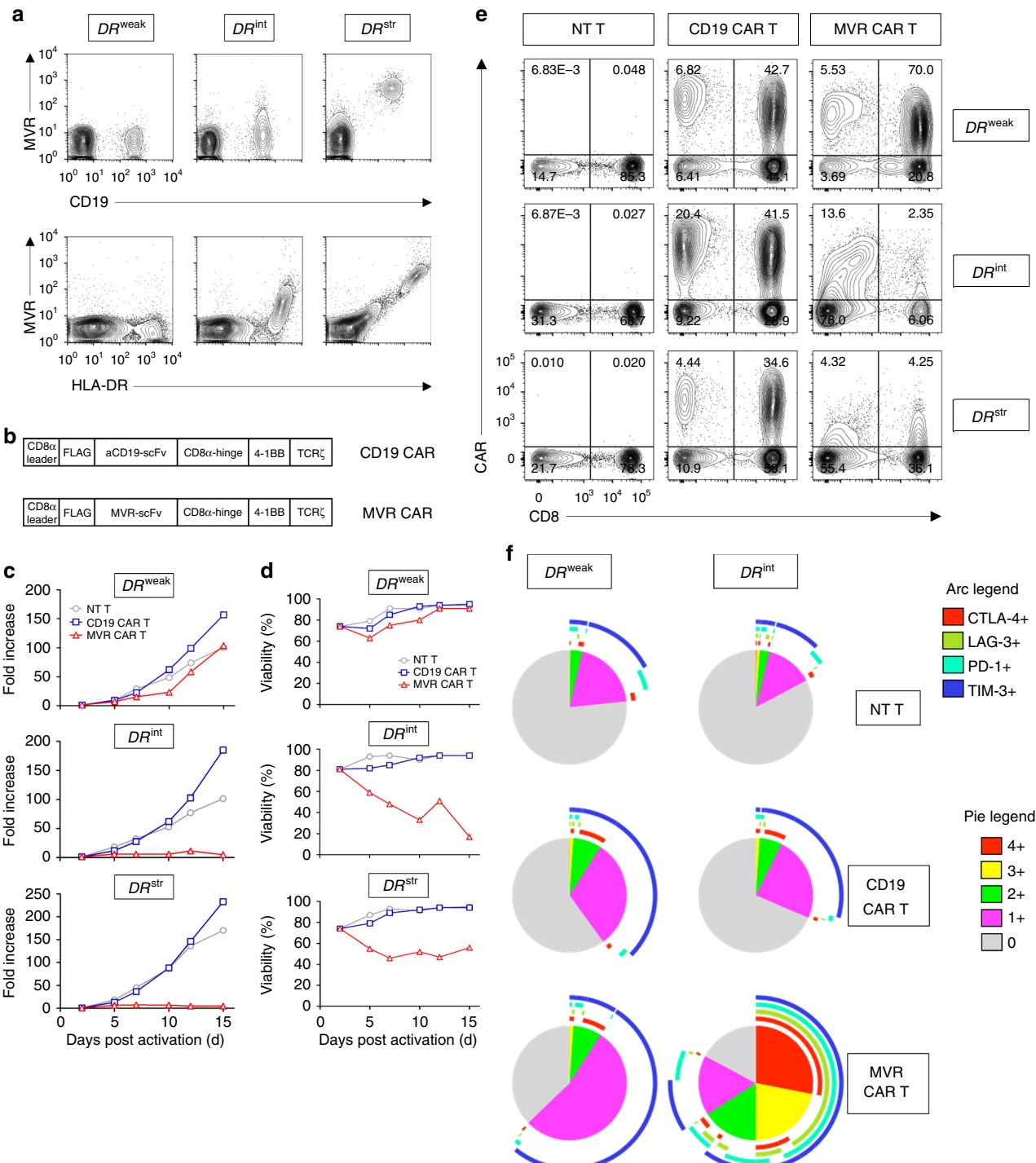

**Fig. 1** MVR CAR T cells undergo reduced fratricide and exhaustion when their binding to HLA-DR is weak. Cells from donors with different *HLA-DRB1* alleles exhibiting strong, intermediate, or weak binding with MVR (*DR*str, *DR*int, or *DR*weak, respectively; **a**) were used in the following experiments. **a** Binding pattern of the MVR antibody. *DR*str, *DR*int, and *DR*weak peripheral blood mononuclear cells (PBMCs) were analyzed for CD19 and HLA-DR expression. **b** Second-generation CAR constructs designed using anti-CD19 or MVR antibody. **c**, **d** Growth (**c**) and viability (**d**) after transduction of *DR*str, *DR*int, and *DR*weak PBMCs. Fold-increases in cell counts (relative to the number on day 0) and viabilities of non-transduced (NT) T, CD19 CAR T, and MVR CAR T cells were measured at the indicated time points. Both CD19 CAR T and MVR CAR T cells were transduced on day 2. **e** Expression of CAR on NT T, CD19 CAR T, and MVR CAR T cells generated from *DR*str, *DR*int, and *DR*weak PBMCs. Cells were analyzed for CD8 and CAR expression at 13 days post-transduction. **f** Expression of multiple exhaustion markers was assessed in NT T, CD19 CAR T, and MVR CAR T cells generated from *DR*int and *DR*weak PBMCs. The length of the arc of each color indicates the frequency of expression of the corresponding marker. The area of the pie of each color indicates the frequency of cells that express the given number of exhaustion markers. The expression of exhaustion markers in each CAR T cell type was analyzed by gating on CAR-positive cells (Supplementary Fig. 1). Pie charts were drawn based on the flow cytometry data measured in Supplementary Fig. 1 using a SPICE v5.0 (NIAID). **a**, **c**–**f** Representative of two independent experiments

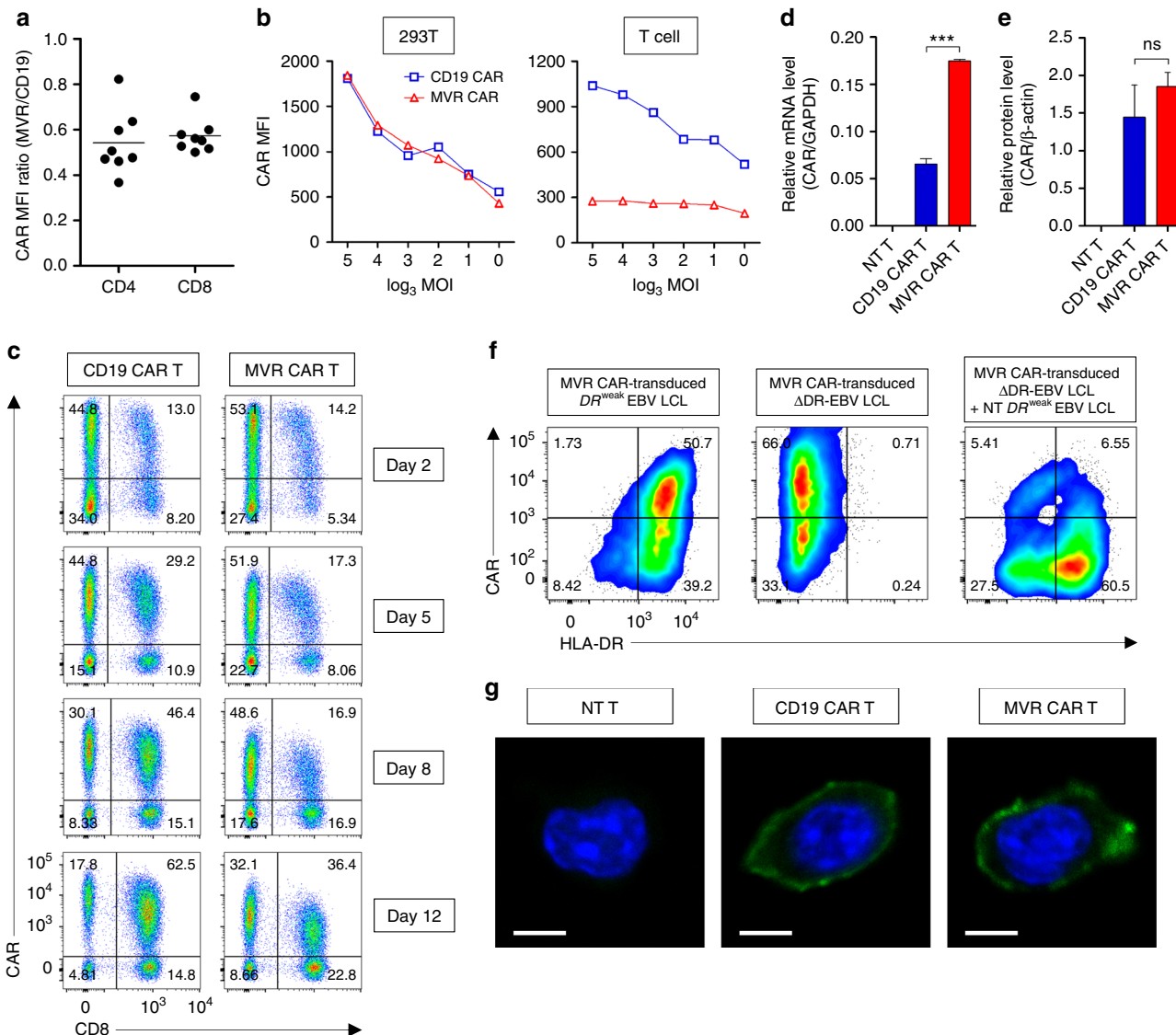

**Fig. 2** Weak MVR CAR–HLA-DR interaction downregulates surface CAR in MVR CAR T cells. Cells from donors with *HLA-DRB1* alleles exhibiting weak binding with MVR (*DR*^weak^; Fig. 1a) were used in the following experiments. **a** Differences in surface CAR expression between CD19 CAR T and MVR CAR T cells. The mean fluorescence intensity (MFI) of the CAR expressed by *DR*^weak^ MVR CAR T cells was divided by that of CD19 CAR T cells. CD4$^+$ or CD8$^+$ T cells were analyzed separately. Flow cytometric data from separately generated CAR T cell preparations was used ($n = 8$). Horizontal lines indicate mean. **b** Lentivirus titer-dependent changes in expression of surface CAR. 293T cells and *DR*^weak^ T cells were transduced with each CAR vector at various multiplicities of infection, and analyzed for MFI of CAR by flow cytometry. 293T cell lines and *DR*^weak^ T cells were analyzed at 5 and 13 days post-transduction, respectively. **c** *DR*^weak^ T cells transduced with the CD19 CAR or MVR CAR vector were analyzed for CAR expression at the indicated times post-transduction. Cells were analyzed for CD8 and CAR expression. **d**, **e** CAR expression analyzed at the mRNA (**d**) and protein (**e**) levels by qPCR and western blotting, respectively. Non-transduced (NT) T, CD19 CAR T, and *DR*^weak^ MVR CAR T cells were subjected to CD4-negative sorting to enrich for CD8$^+$ T cells using CD4 microbeads (130-045-101, Miltenyi Biotec, Inc.) and used for analysis. **f** CAR downregulation in MVR CAR-transduced EBV LCLs; transduced *DR*^weak^ EBV LCLs (left); transduced HLA-DR-defective (ΔDR)-EBV LCLs (center); transduced ΔDR-EBV LCLs co-cultured with non-transduced *DR*^weak^ EBV LCLs for 2 days (right). Cells were analyzed for HLA-DR and CAR expression. **g** Immunofluorescence staining of NT T, CD19 CAR T, and *DR*^weak^ MVR CAR T cells. Representative images from Supplementary Fig. 3 (scale bar = 5 μm). **a** Summary of eight independent experiments. **b–d**, **f** Representative of two independent experiments. **d**, **e** $n = 3$ biological replicates. Mean ± s.e.m. Unpaired two-tailed *t*-test: ns not significant; ***, $p < 0.001$

after contact with *DR*^weak^ EBV LCLs, suggesting that the MVR CAR–HLA-DR interaction is responsible for MVR CAR down-regulation (Fig. 2f). Further immunofluorescence experiments indicated that CAR was localized on the membrane in *DR*^weak^ MVR CAR T cells and CD19 CAR T cells (Fig. 2g and Supplementary Fig. 3). These data suggest that sustained downregulation of surface MVR CAR occurs during in vitro expansion of *DR*^weak^ MVR CAR T cells because of the interaction with HLA-DR.

**MVR CAR T cells kill EBV LCLs while sparing normal B cells.** We showed that *DR*^weak^ MVR CAR T cells survive fratricidal selection and downregulate CAR on their surface. We next investigated the functional consequences of fratricidal selection and CAR downregulation by comparing the immune activation capacity of CD19 CAR T and *DR*^weak^ MVR CAR T cells. We used EBV LCLs continuously expressing CD19 and HLA-DR for activation. To match the *HLA-DRB1* alleles of *DR*^weak^ MVR CAR T cells and target cells, we generated EBV LCLs by EBV

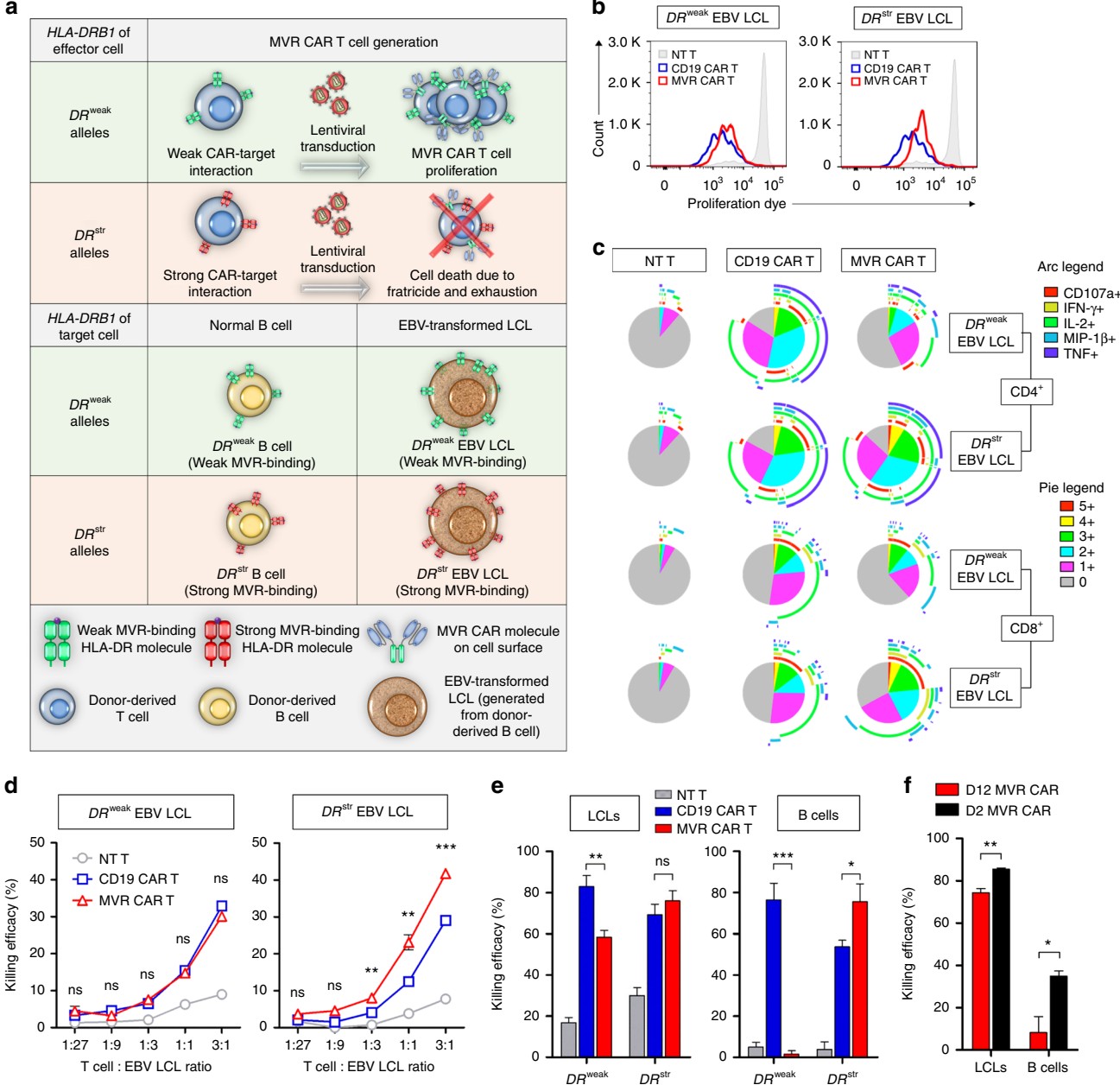

**Fig. 3** MVR CAR T cells have specific killing activity against EBV LCLs. Cells from donors with different *HLA-DRB1* alleles exhibiting strong or weak binding with MVR (*DR*str or *DR*weak, respectively; Fig. 1a) were used in the following experiments. **a** Illustrative summary of MVR CAR T cells and target cells used in this study. **b** Proliferation of T cells measured after activation by *DR*weak EBV LCLs or *DR*str EBV LCLs. **c** Expression of multiple functional markers measured in non-transduced (NT) T, CD19 CAR T, and *DR*weak MVR CAR T cells after contact with *DR*weak or *DR*str EBV LCLs. The length of the arc of each color indicates the frequency of expression of the corresponding marker. The area of the pie of each color indicates the frequency of cells expressing the given number of functional markers. Pie charts were drawn based on the data measured in Supplementary Fig. 4b using a SPICE v5.0 (NIAID). **d** Killing efficacy of NT T, CD19 CAR T, and *DR*weak MVR CAR T cells against *DR*weak or *DR*str EBV LCLs assessed in cytotoxicity assays. **e** Target-specific killing by each CAR T cell type evaluated with an in vitro on-target killing assay. EBV LCLs and peripheral blood mononuclear cells carrying either *DR*weak or *DR*str *HLA-DRB1* alleles were co-incubated with NT T, CD19 CAR T, or *DR*weak MVR CAR T cells. After incubation, the number of viable cells was determined and the killing efficacy was calculated as indicated in Supplementary Fig. 5a. **f** Target-specific killing of *DR*weak MVR CAR T cells on day 2 or 12 post-transduction (D2 or D12, respectively). *DR*weak EBV LCLs were co-incubated with D2 or D12 MVR CAR T cells. After incubation, the number of viable cells was determined and killing efficacy was calculated as indicated in Supplementary Fig. 5b. **b**, **c**, **e** Representative of three independent experiments. **d**, **f** Representative of two independent experiments. **d**–**f** $n = 3$ biological replicates. Mean ± s.e.m. Unpaired two-tailed *t*-test: ns not significant; *$p < 0.05$; **$p < 0.01$; ***$p < 0.001$

transformation of *DR*weak B cells. Accordingly, we compared the functional activities of CD19 CAR T and *DR*weak MVR CAR T cells against *DR*weak EBV LCLs (Fig. 3a). *DR*str EBV LCLs, whose HLA-DRs bind strongly to MVR CAR and hence induce strong immune activation, served as positive controls.

First, we assessed proliferation, a typical feature of T cell activation. *DR*weak MVR CAR T cells multiplied following contact with *DR*weak EBV LCLs, indicating that they were activated by EBV LCLs (Fig. 3b). Next, we investigated polyfunctionality, i.e., simultaneous degranulation and cytokine and/or chemokine

secretion (characterized by expression of CD107a, IFN-γ, IL-2, MIP-1β, and TNF; Supplementary Fig. 4a), by which T cells directly suppress tumors and activate the immune system[21–23]. $DR^{weak}$ MVR CAR T cells exhibited lower polyfunctionality than CD19 CAR T cells in response to $DR^{weak}$ EBV LCLs, and the difference was greater for CD4+ T cells than for CD8+ T cells (Fig. 3c and Supplementary Fig. 4b). In contrast, $DR^{str}$ EBV LCLs induced a greater polyfunctional response in $DR^{weak}$ MVR CAR T cells than in CD19 CAR T cells, suggesting that the reduced polyfunctionality in response to $DR^{weak}$ EBV LCLs was not due to a reduced capacity of $DR^{weak}$ MVR CAR T cells but to the weak

activation signal induced by the interaction between $DR^{weak}$ HLA-DR and MVR CAR.

An important function of CAR T cells is to induce the cell death of target cells. We assessed the cytotoxic killing efficacy of $DR^{weak}$ MVR CAR T cells against EBV LCLs. $DR^{weak}$ MVR CAR T cells exhibited dose-dependent killing of $DR^{weak}$ EBV LCLs similar to the killing by CD19 CAR T cells, whereas they killed $DR^{str}$ EBV LCLs more efficiently than CD19 CAR T cells (Fig. 3d). Based on the limited fratricide observed during initial expansion of $DR^{weak}$ MVR CAR T cells (Fig. 1c, d), these results indicate that the low affinity between $DR^{weak}$ HLA-DR and MVR

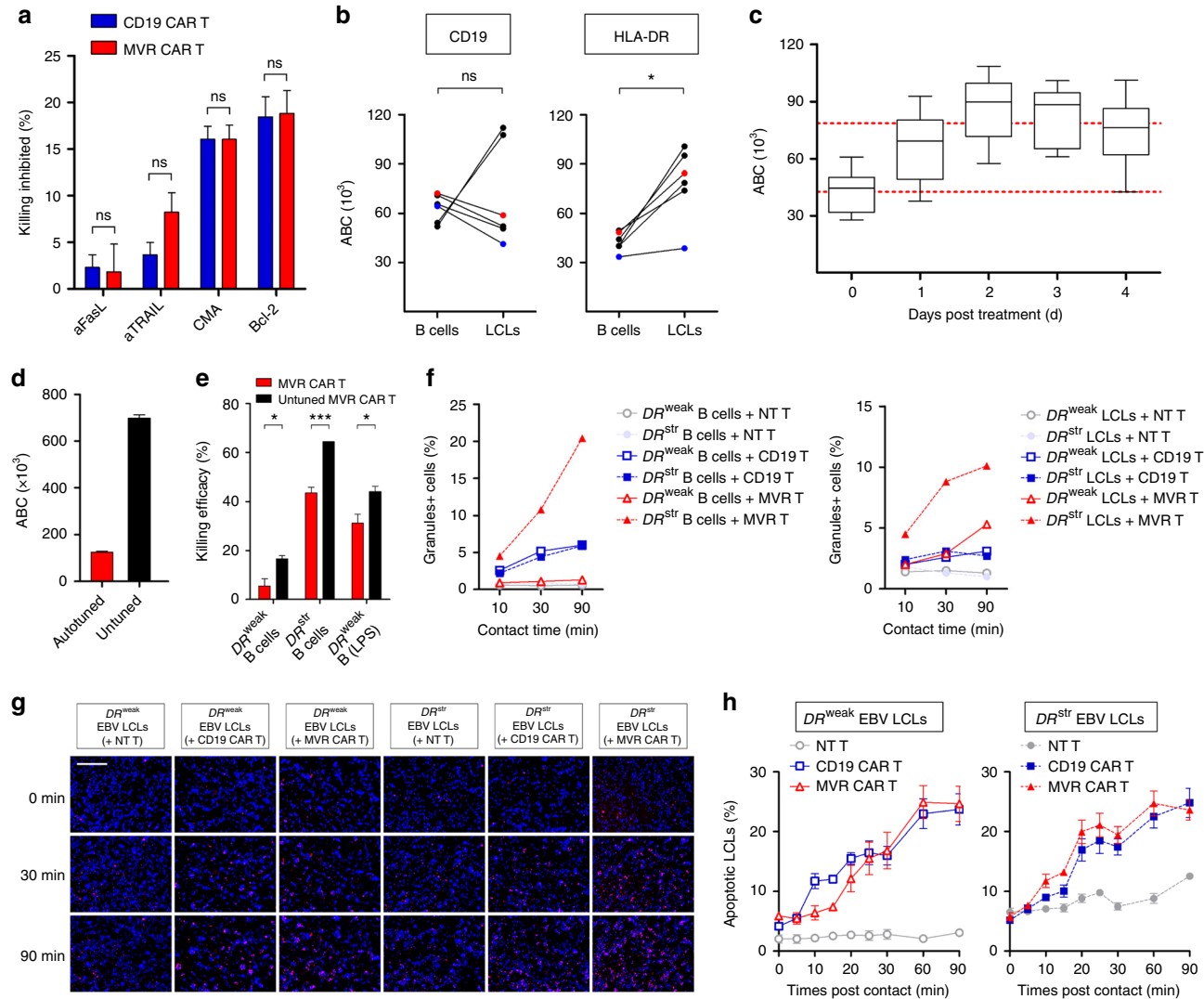

**Fig. 4** MVR CAR T cells selectively kill HLA-DR-upregulated EBV LCLs by sensing antigen level. Cells from donors with different *HLA-DRB1* alleles exhibiting strong or weak binding to MVR ($DR^{str}$ or $DR^{weak}$, respectively; Fig. 1a) were used as follows. **a** Killing inhibition in the presence of the indicated blocking agents. $DR^{weak}$ EBV LCL killing by each CAR T cell type was measured as in Fig. 3f ($n = 3$). **b** Expression of CD19 and HLA-DR on B cells and EBV LCLs. Antibody binding capacity (ABC) is an index of numbers of target molecules. Dots connected by the same line involve the same donor ($n = 6$). Red and blue dots indicate $DR^{weak}$ and $DR^{str}$ cells, respectively. **c** HLA-DR expression on lipopolysaccharide-stimulated B cells. Cells were analyzed as in **b** for 4 days ($n = 6$). Upper and lower dotted lines indicate average HLA-DR levels of EBV LCLs and B cells, respectively, measured in **b**. Whiskers indicate minimum and maximum values. **d** MVR CAR expression on the surface of untuned and autotuned $DR^{weak}$ MVR CAR T cells (day 2 and 12 in Fig. 2c) evaluated as in **b** ($n = 4$). **e** Killing efficacies of untuned and tuned $DR^{weak}$ MVR CAR T cells evaluated as in Fig. 3f. $DR^{weak}$ B cells, $DR^{str}$ B cells, and $DR^{weak}$ B cells treated with lipopolysaccharide for 3 days were used as target cells ($n = 3$). **f** Proportions of B cells and EBV LCLs containing transferred granules after contact with T cells. Cells contacted for indicated times were analyzed as shown in Supplementary Fig. 6. NT non-transduced. **g**, **h** Time-lapse analysis of apoptotic EBV LCLs after contact with T cells. **g** EBV LCLs (blue) undergoing apoptosis (red) identified by detecting magenta color (scale bar indicates 250 μm). **h** Proportions of apoptotic EBV LCLs at indicated time points. Three different areas of each sample were analyzed. **a** Representative of three independent experiments. **f**, **h** Representative of two independent experiments. **a**, **d**, **e** *n* indicates biological replicates. **a**, **d**, **e**, **h** Mean ± s.e.m. **b** Two-tailed Wilcoxon matched pairs test. **a**, **e** Unpaired two-tailed *t*-test; ns not significant; *$p < 0.05$; ***$p < 0.001$

CAR can be used to distinguish EBV LCLs from activated T cells, although both express $DR^{weak}$ HLA-DR.

CD19 CAR T cells cause on-target off-tumor toxicity such as B cell aplasia in CD19 CAR T cell-infused patients[1,24,25]. To assess the on-target off-tumor killing efficacy of $DR^{weak}$ MVR CAR T cells, we designed an in vitro on-target killing assay to evaluate cytotoxicity against B cells and EBV LCLs simultaneously (Supplementary Fig. 5a). In agreement with their killing efficacies,

CD19 CAR T and $DR^{weak}$ MVR CAR T cells showed cytotoxic activity against $DR^{str}$ and $DR^{weak}$ EBV LCLs (Fig. 3e). Strikingly, $DR^{weak}$ B cells were not affected by $DR^{weak}$ MVR CAR T cells, whereas $DR^{str}$ B cells were killed. To determine whether fratricidal selection and CAR downregulation affected the killing selectivity of $DR^{weak}$ MVR CAR T cells, we subjected $DR^{weak}$ MVR CAR T cells on day 2 and day 12 post-transduction (D2 and D12 MVR CAR T, respectively, in Fig. 2c) to an in vitro on-

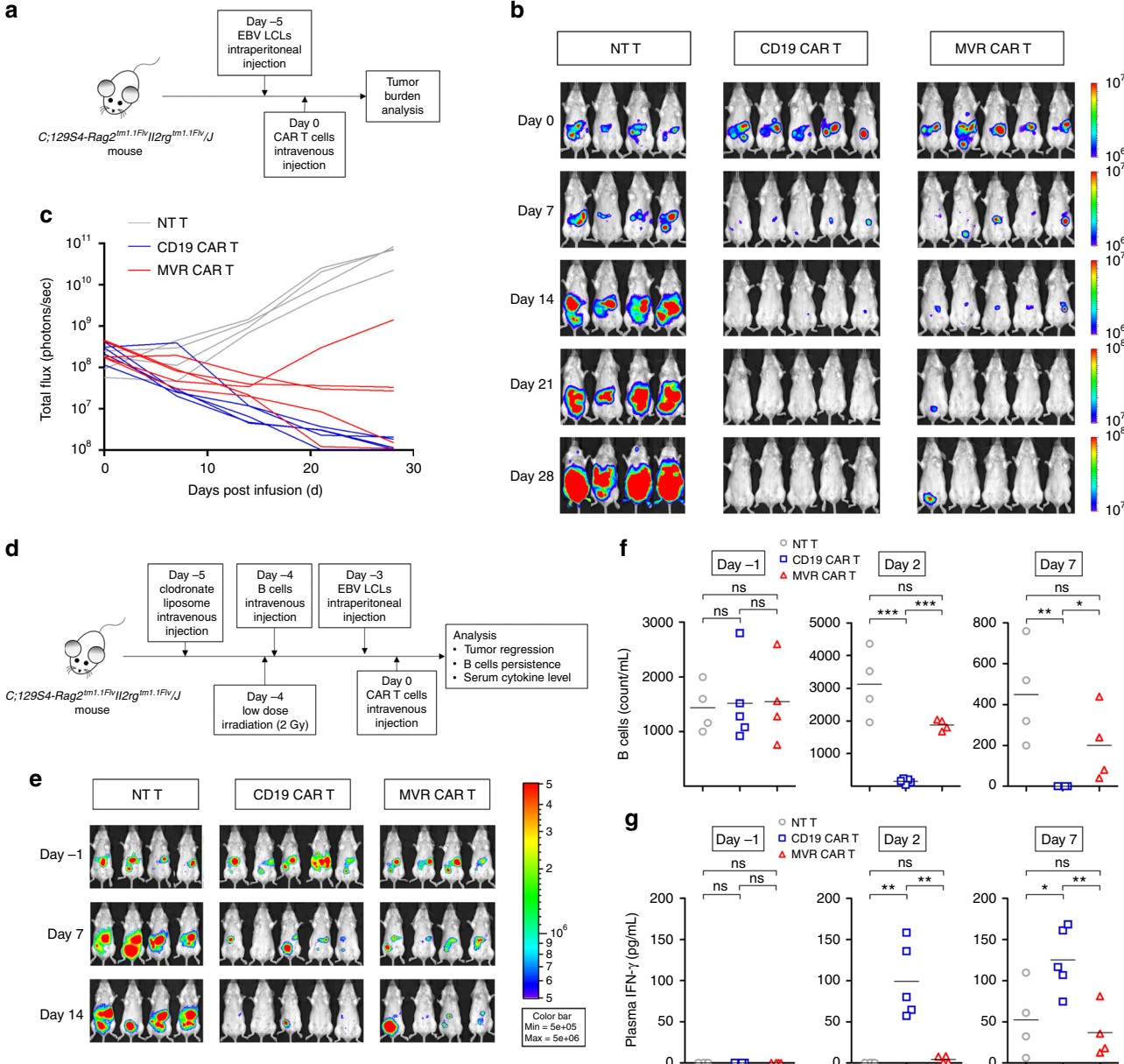

**Fig. 5** Autotuned MVR CAR T cells sense antigen levels in vivo. EBV LCL-targeting specificity of MVR CAR T cells was verified in vivo. T cells, B cells, EBV LCLs from donors with *HLA-DRB1* alleles exhibiting weak binding with MVR ($DR^{weak}$; Fig. 1a) were used in the following experiments. **a–c** Efficacies of $DR^{weak}$ EBV LCL suppression after infusion with non-transduced (NT) T, CD19 CAR T, or $DR^{weak}$ MVR CAR T cells. Luciferase activity in mice grafted with luciferase-labeled $DR^{weak}$ EBV LCLs was measured on 0, 7, 14, 21, and 28 days post-T cell infusion. **a** Procedure for evaluating EBV LCL suppression in vivo. Images (**b**) and total luminescence (**c**) were collected and analyzed. **d** Procedure of in vivo on-target killing assay. Xenografting of $DR^{weak}$ B cell/$DR^{weak}$ EBV LCL was followed by infusion with NT T, CD19 CAR T, or $DR^{weak}$ MVR CAR T cells, and subsequent efficacy analysis. **e** Efficacy of EBV LCL suppression after infusion with each T cell observed for 14 days. Luciferase activity in mice grafted with $DR^{weak}$ B cells and luciferase-labeled $DR^{weak}$ EBV LCLs was measured on −1, 7, and 14 days post-T cell infusion. **f** B cell persistence in T cell-infused mice on −1, 2, and 7 days post-T cell infusion. Peripheral blood of each mouse was stained with a panel of antibodies and analyzed as described in Supplementary Fig. 7. **g** Plasma IFN-γ levels measured in mice infused with NT T, CD19 CAR T, or $DR^{weak}$ MVR CAR T cell on −1, 2, and 7 days post-T cell infusion. **a–c** NT T, $n = 4$ mice; CD19 CAR T, $n = 5$ mice; MVR CAR T, $n = 5$ mice. **d–g** NT T, $n = 4$ mice; CD19 CAR T, $n = 5$ mice; MVR CAR T, $n = 4$ mice. **f, g** Horizontal lines indicate mean. **f, g** Unpaired two-tailed $t$-test: ns not significant; *$p < 0.05$; **$p < 0.01$; ***$p < 0.001$

target killing assay. D2 MVR CAR T cells exhibited significantly higher killing activity than D12 (unpaired two-tailed $t$-test; LCLs, $p = 0.0050$; B cells, $p = 0.0285$; Fig. 3f) against both $DR^{weak}$ B cells and $DR^{weak}$ EBV LCLs, indicating that fratricidal selection and CAR downregulation modulated the cytotoxicity threshold. Taken together, these observations suggest that $DR^{weak}$ MVR CAR T cells are activated by $DR^{weak}$ EBV LCLs and exclusively kill $DR^{weak}$ EBV LCLs; this killing is further improved by downregulation of MVR CAR. As downregulation of surface CAR occurs autonomously during fratricidal selection and eventually results in sensitivity tuning, we named the process as 'autotuning'.

**Specific targeting depends on antigen and CAR levels**. $DR^{weak}$ B cells were more susceptible to cell death when co-cultured with D2 (untuned) MVR CAR T cells than with D12 (autotuned) cells (Fig. 3f). The extent of cell death, however, was still lower than that of $DR^{weak}$ EBV LCLs. This indicates that another factor makes $DR^{weak}$ EBV LCLs more susceptible to cytotoxicity induced by $DR^{weak}$ MVR CAR T cells. One possible factor is the presence of death receptors, as EBV LCLs express Fas and TRAIL-R2, which induce cell death after binding to FasL and TRAIL[26]. To analyze this effect, we blocked the four major pathways of cytotoxic killing (FasL, TRAIL, perforin-1, and granzyme B)[27] and evaluated the killing efficacy of CAR T cells. Inhibition of killing by blocking agents did not differ between $DR^{weak}$ MVR CAR T cells and CD19 CAR T cells. Blocking of FasL and TRAIL had little or no effect on killing efficacy, while inhibition of perforin-1 or granzyme B reduced killing efficacy by 15–20% (Fig. 4a). This suggests that the cell death of $DR^{weak}$ EBV LCLs mainly involves the cytolytic granule-mediated pathway, but not death receptor-mediated pathways.

Another possible factor that makes $DR^{weak}$ EBV LCLs more susceptible to cytotoxic killing is upregulation of HLA-DR[28], as an increased level of the target antigen results in more efficient killing by CAR T cells[10,29]. Therefore, we investigated changes in the expression of CD19 and HLA-DR on the surface of B cells and EBV LCLs. HLA-DR was upregulated in all tested donors after transformation with EBV (B cells = 42,590 ± 2458, EBV LCLs = 78,513 ± 8963, mean ± s.e.m., $n = 6$), whereas CD19 was downregulated in four donors and upregulated in only two donors (Fig. 4b). To examine the contribution of $DR^{weak}$ HLA-DR upregulation to the $DR^{weak}$ EBV LCL-specific killing of $DR^{weak}$ MVR CAR T cells, we assessed the susceptibility to killing of $DR^{weak}$ HLA-DR-upregulated B cells. B cells present in lipopolysaccharide-stimulated peripheral blood mononuclear cells (PBMCs) expressed higher levels of HLA-DR than those in unstimulated PBMCs (Fig. 4c). HLA-DR expression on B cells peaked at 2–3 days post-stimulation, and the peak level was similar to that on EBV LCLs (lipopolysaccharide-stimulated B cells on day 2 = 86,383 ± 7217, day 3 = 82,945 ± 6395, mean ± s.e.m., $n = 6$). We used $DR^{weak}$ PBMCs stimulated with lipopolysaccharide for 3 days as target cells in a killing assay, as well as autotuned and untuned MVR CAR T cells (with a 5.6-fold difference in CAR expression) as effector cells (Fig. 4d; autotuned = 124,854 ± 2531, untuned = 698,123 ± 7458, mean ± s.e.m., $n = 4$). Lipopolysaccharide-stimulated $DR^{weak}$ B cells were more susceptible to $DR^{weak}$ MVR CAR T cell-induced killing than unstimulated $DR^{weak}$ B cells (Fig. 4e). Moreover, untuned $DR^{weak}$ MVR CAR T cells were more efficient at killing than autotuned cells. These observations indicate that both autotuning and HLA-DR upregulation contribute to increased cytotoxic killing.

Strong TCR signals induce active granule transfer from T cells to target cells[30,31]. Therefore, the extent of transfer of granules by MVR CAR may depend on the strength of the MVR CAR–HLA-

DR interaction. We measured the quantity of granules transferred over time after contact between CAR T cells and B cells or EBV LCLs (Supplementary Fig. 6a). There was no measurable granule influx into $DR^{weak}$ B cells for 90 min after contact with $DR^{weak}$ MVR CAR T cells, whereas granule influx into $DR^{weak}$ EBV LCLs was easily detected and increased over time. In contrast, granule influx into $DR^{str}$ B cells and $DR^{str}$ EBV LCLs was rapid after contact with $DR^{weak}$ MVR CAR T cells and was 2- to 4-fold greater than with CD19 CAR T cells (Fig. 4f and Supplementary Fig. 6b, c).

Lytic granules transferred from T cells actively induce apoptosis of target cells[32]. Time-lapse imaging of caspase 3/7-activated EBV LCLs in contact with CAR T cells revealed that CD19 CAR T and $DR^{weak}$ MVR CAR T cells progressively increased the proportion of apoptotic $DR^{str}$ and $DR^{weak}$ EBV LCLs (Fig. 4g, h and Supplementary Movie 1). The kinetics of the interactions were similar to those of granzyme influx, suggesting that granule transfer was the main cause of $DR^{weak}$ MVR CAR T cell-induced cytotoxicity, as indicated by the results in Fig. 4a. Collectively, these data suggest that autotuned $DR^{weak}$ MVR CAR T cells sense the level of $DR^{weak}$ HLA-DR and induce the death of target cells by lytic granule transfer.

**MVR CAR T cells sense enhanced HLA-DR level in vivo**. Finally, we evaluated $DR^{weak}$ MVR CAR T cells in an animal model. The transfer of $DR^{weak}$ MVR CAR T cells into $DR^{weak}$ EBV LCL-xenograft $C;129S4$-$Rag2^{tm1.1Flv}Il2rg^{tm1.1Flv}/J$ mice resulted in suppression of EBV LCL-induced tumors (Fig. 5a, b). The efficacy appeared higher for CD19 CAR T cells than for $DR^{weak}$ MVR CAR T cells, although the difference was not significant (two-way ANOVA; $p = 0.5175$; Fig. 5c). To confirm the antigen-quantity-based target-cell selectivity of $DR^{weak}$ MVR CAR T cells under physiological conditions, we designed an in vivo on-target killing assay. In this assay, we used mice grafted with $DR^{weak}$ B cells and $DR^{weak}$ EBV LCLs. This enabled observation of the rate of eradication of the two cell populations in CAR T cell-infused mice (Fig. 5d). As expected, tumor regression was observed in mice infused with $DR^{weak}$ MVR CAR T cells or CD19 CAR T cells, but not in those infused with NT T cells (Fig. 5e). Notably, peripheral blood $DR^{weak}$ B cells persisted in $DR^{weak}$ MVR CAR T cell-infused mice, whereas most $DR^{weak}$ B cells were eliminated within 2 days in CD19 CAR T cell-infused mice (Fig. 5f and Supplementary Fig. 7a, b). We observed a difference in the $DR^{weak}$ B cell count between mice infused with $DR^{weak}$ MVR CAR T cells and those infused with CD19 CAR T cells until 7 days post-T cell infusion, when tumor suppression was active. Interestingly, the expression of HLA-DR by residual $DR^{weak}$ B cells from $DR^{weak}$ MVR CAR T cell-infused mice was lower than by $DR^{weak}$ B cells from NT T cell-infused mice (Supplementary Fig. 7c), suggesting that, as observed in vitro (Fig. 4c,e), HLA-DR-upregulated $DR^{weak}$ B cells activated by xeno-reaction had increased susceptibility to $DR^{weak}$ MVR CAR T cell-induced cytotoxicity in vivo. In addition, the plasma IFN-γ level of the $DR^{weak}$ MVR CAR T cell-infused mice was lower than that of the CD19 CAR T cell-infused mice (Fig. 5g), in agreement with the in vitro result (Fig. 3c and Supplementary Fig. 4b). Together, these data confirm the in vitro results showing that $DR^{weak}$ MVR CAR T cells sense $DR^{weak}$ HLA-DR levels under physiological conditions.

**Discussion**
During thymic development, T cells undergo TCR-affinity-dependent negative and positive selection[33]. This is crucial for tuning T cell sensitivity so that T cells protect against pathogens and malignant cells, but are not auto-reactive[34–37]. CAR T cells,

however, do not undergo sensitivity selection, and this tolerance confers unprecedented therapeutic activity against cancers while also potentially causing serious side effects[25,38–40]. In the current study, sensitivity selection was mimicked by fratricide. $DR^{str}$ and $DR^{int}$ MVR CAR T cells were involved in substantial fratricide, as the affinity between the MVR CAR and the HLA-DRs was sufficiently high to induce strong immune activation. Intense immune activation was inferred from the elevated exhaustion level of $DR^{int}$ MVR CAR T cells (Fig. 1f and Supplementary Fig. 1). In contrast, $DR^{weak}$ MVR CAR T cells exhibited mild fratricide and exhaustion, indicating that the affinity between MVR CAR and $DR^{weak}$ HLA-DR was sufficiently low to limit the immune response. Indeed, $DR^{weak}$ MVR CAR T cells were not cytotoxic to $DR^{weak}$ B cells, while they killed $DR^{str}$ B cells. Fratricidal selection is a useful strategy for CAR T cell development in which potentially harmful CAR T cells are detected and removed.

In the current study, cell surface CAR was downregulated on autotuned $DR^{weak}$ MVR CAR T cells through a weak interaction between MVR CAR and $DR^{weak}$ HLA-DR, and downregulation was sustained for at least two activation cycles (>4 weeks; Supplementary Fig. 8). Furthermore, autotuned $DR^{weak}$ MVR CAR T cells showed lower cytokine levels and a higher immune activation threshold. These characteristics are similar to those of T cells that have undergone long-term downregulation of the TCR[12]. Based on the synthetic nature of CARs, it is interesting that CAR T cells regulate their functional thresholds by direct tuning of the CAR, which also occurs in long-term TCR downregulation in normal T cells. This indicates that a shared pathway underlies the long-term downregulation of TCRs and CARs; however, further studies are required to elucidate the posttranslational mechanisms of these effects.

The relationship between T cell activity and the levels of CAR and antigen has been evaluated in previous studies. Fedorov et al.[41] and Chang et al.[42] demonstrated that CAR T cells sorted for higher CAR expression had higher effector functions than those sorted for low CAR expression, and Fedorov et al.[41] and Han et al.[43] reported that the antigen expression level and CAR T cell activity were correlated. This correlation was more significant when CAR T cells and blocking antibody were used together[44] and when low affinity CARs were used[10,29]. Our results are significant because they demonstrate two characteristics of the control of target cell selectivity derived from the special properties of HLA-DR-specific MVR CAR. First, we detected the existence of an autonomous desensitization mechanism (autotuning) that modulates CAR T cell reactivity. $DR^{weak}$ MVR CAR T cells recognized HLA-DR on the T cell surface and induced sustained CAR downregulation without severe fratricide and exhaustion. In contrast to the results of studies of Fedorov et al.[41] and Chang et al.[42], where low CAR-expressing T cells were sorted based on their CAR expression levels, we identified and investigated this CAR downregulation effect in which CAR T cells spontaneously modulate their reactivity based on T cell-intrinsic characteristics. This result reveals the presence of self-limiting behavior of CAR T cells in the form of autotuning. Further studies examining the condition in which CAR expression is optimally downregulated should be conducted to develop safer CAR T cells. Furthermore, the current study demonstrated the advantage of combining a low-affinity antigen with low CAR expression, thus improving target cell selectivity. Previous studies also reported that low CAR expression significantly decreased CAR T cell activity in target cells[41,42] and low CAR affinity altered CAR T cell function[10,29]. Because autotuned $DR^{weak}$ MVR CAR T cells have both low affinity and low CAR expression, they exhibited much greater target cell selectivity based on antigen levels. Particularly, $DR^{weak}$ MVR CAR T cells efficiently discriminated between primary B

cells and EBV-transformed B cells in in vitro and in vivo (Figs. 3e, 4f, 5f). Notably, the low affinity and low CAR expression were both important, as $DR^{weak}$ B cells were killed by untuned $DR^{weak}$ MVR CAR T cells (low affinity only; Fig. 4e) and $DR^{str}$ B cells were killed by autotuned $DR^{weak}$ MVR CAR T cells (low CAR expression only; Fig. 4e).

In terms of therapeutic application, this study suggests the advantage of MVR CAR T cell therapy for HLA-DR-increased B cell malignancies. Because HLA-DR was upregulated in EBV-transformed B cell lines used in this study[28], $DR^{weak}$ MVR CAR T cells can efficiently kill HLA-DR-upregulated $DR^{weak}$ EBV-transformed B cells. Based on the link between EBV and B cell lymphomagenesis[45], surface HLA-DR expression may be increased in B cell lymphoma. Indeed, some representative malignant B cell lines show similar or higher levels of surface HLA-DR expression than the EBV-transformed B cells used in the current study (Fig. 4b and Supplementary Fig. 9). This suggests the therapeutic potential of MVR CAR T cells in HLA-DR-increased B cell malignancies with reduced on-target off-tumor side effects, as observed in the EBV-transformed lymphoma model. One factor limiting MVR CAR T cell therapy is the limited frequency of $DR^{weak}$ HLA-DRB1 alleles, which reduces the relevant patient pool. Further reverse engineering of an MVR antibody to reduce its affinity for $DR^{str}$ and $DR^{int}$ HLA-DRB1 alleles may expand its applicability.

The avidity of T cells can be assessed on a scale that integrates the expression level of the receptor and receptor–antigen affinity[46]. T cell avidity determines the minimum antigen level above which TCR–antigen complexes form clusters that eventually lead to T cell activation through an immunological synapse[47–49]. Taylor et al. investigated the quantitative requirement for T cell signaling by designing a DNA-CAR system[50]. Importantly, the authors performed stochastic simulations based on their results and found that under some conditions, ~3-fold changes in ligand density or affinity resulted in a sharp transition to receptor clustering that led to T cell signaling and function. Consistent with this finding, we observed that ~2-fold changes in HLA-DR expression in target cells crossed the threshold for induction of cytotoxic killing by CAR T cells (Fig. 4c, e). Furthermore, we showed that ~6-fold changes in CAR expression enhanced MVR CAR–HLA-DR interaction and resulted in target cell killing (Fig. 4d, e). The acute sensitivity to the levels of HLA-DR and MVR CAR depends on low affinity, as $DR^{str}$ HLA-DR-expressing cells were killed by $DR^{weak}$ MVR CAR T cells regardless of HLA-DR or CAR expression levels (Figs. 3e, 4e). Thus, together, autotuning, low affinity, and HLA-DR-upregulation tuned the avidity and reactivity of $DR^{weak}$ MVR CAR T cells, resulting in accurate target-cell selectivity (Supplementary Fig. 10).

In conclusion, the current study highlights the requirements for minimizing unwanted killing by CAR T cells under physiological conditions. These requirements are weak affinity, selection of an antigen whose expression level is increased by >2-fold in malignant cells, and modulation of CAR expression level by an efficient process such as autotuning. These findings advance the understanding of the characteristics of CAR T cells.

## Methods

**Plasmid design**. A DNA construct encoding the single-chain variable fragment (scFv) form of the MVR antibody[51] was generated by connecting the VL and VH regions with a GS linker using standard DNA cloning techniques (Supplementary Table 1). A CD8α leader sequence was inserted at the 5′-terminal of the MVR-scFv sequence to allow the protein to be secreted (Supplementary Table 1). For easier purification and detection, His-tag and FLAG-tag sequences were attached at the 5′- and 3′-terminals of the MVR-scFv sequence, respectively (Supplementary Table 1). MVR-scFv was then cloned into a pcDNA3.1(+) expression vector (V790-20, Invitrogen, Carlsbad, CA, USA) to generate pcDNA3.1–MVR-scFv. To create the MVR CAR construct, the MVR-scFv sequence was grafted into the previously

described lentiviral vector pELPS-19BBz, which encodes a second-generation CD19 CAR construct[52,53], using standard DNA cloning techniques. The FLAG-tag sequence was inserted between the CD8α leader and scFv sequences of CD19 CAR and MVR CAR to generate pELPS-FLAG19BBz and pELPS-FLAGMVRBBz, respectively (Fig. 1b), so that expression of each construct could be detected in an unbiased manner with an anti-FLAG antibody. To generate pLCv2-DRB1, an HLA-DRB1-targeting sgRNA/Cas9 expression vector, the HLA-DRB1 exon3-targeting spacer sequence, was inserted into lentiCRISPRv2 (52961, Addgene, Cambridge, MA, USA) using standard DNA cloning techniques (Supplementary Table 1).

**Cells and media**. PBMCs were obtained with informed consent from healthy volunteer donors at the National Cancer Center Research Institute using a National Cancer Center Institutional Review Board-approved protocol. PBMCs were isolated by density gradient centrifugation and either used immediately or stored in liquid nitrogen. EBV LCLs were generated from PBMCs by transformation with EBV. In detail, exponentially growing B95-8 cells were incubated for 3 days at 37 °C. The supernatant was filtered through a 0.45-μm filter and used for transformation. For EBV-transformation, $10^7$ PBMCs in 2.5 mL media were mixed with 2.5 mL of EBV-containing supernatant and incubated for 2 h at 37 °C. The mixed cells were transferred to a T75 flask, and 5 mL of media containing 1 μg/mL cyclosporine A was added. After 3 weeks of incubation, the outgrowing immortalized B cells were checked for CD19 and HLA-DR expression and used in this study. The EBV LCL-lucH cell line was generated by single-cell cloning after electroporation of $DR^{weak}$ EBV LCLs in the presence of the pGL4.51 vector (E132A, Promega, Madison, WI, USA). ΔDR-EBV LCL, which has a defective HLA-DR molecule, was generated by introducing pLCv2-DRB1 into $DR^{weak}$ EBV LCLs by electroporation. For electroporation, cells and plasmids were placed in 4-mm cuvettes and pulsed at 250 V, 975 μF with a Gene Pulser Xcell electroporation system (Bio-Rad Laboratories, Inc., Hercules, CA, USA) using the exponential decay program. After electroporation, HLA-DR-negative $DR^{weak}$ EBV LCLs were sorted with a FACSAria flow cytometer (BD Biosciences, Franklin Lakes, NJ, USA). 1A2 (CRL-8119, ATCC, Manassas, VA, USA), BC-1 (CRL-2230, ATCC), JVM-2 (CRL-3002, ATCC), Daudi (CCL-213, ATCC), Raji (CCL-86, ATCC), Ramos (CRL-1596, ATCC), NALM6 (CRL-3273, ATCC), B95-8 (CRL-1612, ATCC), EBV LCLs, EBV LCLs-lucH, and ΔDR-EBV LCLs were cultured in RPMI 1640 (LM011-01, Welgene, Inc., Daejeon, Korea) supplemented with 1% penicillin/streptomycin (15140-122, Gibco, Grand Island, NY, USA) and 10% heat-inactivated fetal bovine serum (FBS-BBT-5XM, Rocky Mountain Biologicals, Inc., Missoula, MT, USA). Expanded T cells and PBMCs were cultured in RPMI 1640 (LM011-77, Welgene, Inc.) supplemented with 1% penicillin/streptomycin (15140-122, Gibco) and 10% heat-inactivated fetal bovine serum (FBS-BBT-5XM, Rocky Mountain Biologicals, Inc.). Lenti-X 293T (632180, Clontech Laboratories, Inc., Mountain View, CA, USA) and 293T cell lines were cultured in DMEM (LM001-05, Welgene, Inc.) supplemented with 1% penicillin/streptomycin (15140–122, Gibco) and 10% heat-inactivated fetal bovine serum (FBS-BBT-5XM, Rocky Mountain Biologicals, Inc.). All cell lines used in this study were cultured in the presence of ZellShield (13-0050, Minerva Biolabs, Hackensack, NJ, USA) within the past year, and validated using an e-Myco VALiD Mycoplasma PCR Detection Kit (S25239, iNtRON Biotechnology, Inc., Seoul, Korea) to be free from mycoplasma. Cell line authentication was not conducted.

**MVR-scFv production**. To produce purified MVR-scFv protein, pcDNA3.1–MVR-scFv was transfected into 293T cells. MVR-scFv protein secreted into the supernatant was collected at 48 h post-transfection and purified with a Ni-NTA Purification System (R901-10, Thermo Fisher Scientific, Inc., Waltham, MA, USA) according to the manufacturer's protocol.

**Flow cytometry and antibodies**. To analyze the expression of surface markers, $1 \times 10^6$ cells were stained with specific antibodies for 30 min at 4 °C. To assess the binding of MVR-scFv to surface receptors, $1 \times 10^6$ cells were stained with 1 μg of purified MVR-scFv for 30 min at 4 °C, washed once, and stained with PE- or APC-conjugated anti-FLAG antibody for 30 min at 4 °C. The cells were washed twice and fixed with 1% paraformaldehyde before analysis. To analyze intracellular antigens, cells were stained with intracellular antigen-specific antibodies using a Cytofix/Cytoperm Fixation/Permeabilization Kit (554714, BD Biosciences). To evaluate proliferation after target antigen contact, T cells were labeled with a CellTrace violet cell proliferation kit (C34557, Thermo Fisher Scientific, Inc.) and EBV LCLs were γ-irradiated at a dose of 30 Gy using a Gammacell 3000 $^{137}$Cs irradiator (Best Theratronics, Ltd., Ontario, Canada). A total of $1.2 \times 10^6$ cells were then mixed at a T cell:EBV LCL ratio of 3:1 and cultured for 5 days in the presence of 200 IU/mL of human recombinant IL-2. On day 5, the cultured cells were washed twice and fixed with 1% paraformaldehyde before analysis. Polyfunctionality was evaluated by measuring the levels of CD107a, IFN-γ, IL-2, MIP-1β, and TNF. EBV LCLs were labeled with a CellTrace carboxyfluorescein succinimidyl ester cell proliferation kit (C34554, Thermo Fisher Scientific, Inc.) and used to activate T cells. A total of $1.2 \times 10^6$ cells were co-incubated at a T cell:EBV LCL ratio of 3:1 for 6 h in 48-well plates in the presence of a protein transport inhibitor cocktail (00-4980, Thermo Fisher Scientific, Inc.) and CD107a-specific

antibody. The cells were stained with anti-CD4 antibody, washed twice, and stained intracellularly with IFN-γ-, IL-2-, MIP-1β-, and TNF-specific antibodies. All flow cytometric analysis was performed with FACSCalibur or FACSVerse flow cytometers (BD Biosciences). Further information regarding the antibodies used in this study is shown in Supplementary Table 2.

**Lentivirus preparation**. Lentivirus vectors were generated using Lenti-X 293T packaging cell line and packaging plasmid vectors. On the day before transfection, Lenti-X 293T cells were seeded in a 150-mm culture dish at a density of $10^5$ cells/cm². The next day, on day 0, CAR-encoding lentivirus vector constructs (pELPS-FLAG19BBz and pELPS-FLAGMVRBBz) were transfected into Lenti-X 293T cells with packaging plasmid vectors, pMDLg/pRRE, pRSV-rev, and pMD.G, at a ratio of 16:7:7:1 using Lipofectamine 3000 (L3000075, Thermo Fisher Scientific, Inc.). Supernatants harvested 24 and 48 h post-transfection were concentrated by ultracentrifugation for 90 min at 16,500×g at 4 °C in Thickwall Polyallomer tubes (355642, Beckman Coulter, Inc., Brea, CA, USA). After ultracentrifugation, supernatants were discarded and 1 mL of fresh T cell media was added to each tube. Sealed tubes incubated overnight at 4 °C were filtered through a 0.45-μm filter and aliquoted and stocked at −70 °C until use. Lentivirus titers were determined by calculating transduction units. Human PBMCs were activated using a human T cell activation/expansion kit (130–091–441, Miltenyi Biotec, Inc., Bergisch Gladbach, Germany) on day 0. On day 2, T cells were seeded at a density of $10^5$ cells/well in 96-well flat-bottom plates in the presence of 50 μL T cell media. For transduction, 100 μL of a 3-fold serial-diluted lentivirus vector containing 10 μg/mL of polybrene was added to T cell-seeded wells and spinoculated for 2 h at 1200×g at 25 °C. After spinoculation, the plate was incubated for 2 days at 37 °C, and the transduced T cells were stained with anti-FLAG antibody and analyzed for CAR expression by FACSVerse flow cytometers (BD Biosciences). By determining the dilution rate, which resulted in a transduction rate between 0.05 and 0.1, transduction U/mL of lentivirus was calculated using the following equation: (transduction rate $\times 10^5 \times$ 10)/dilution rate.

**CAR T cell production**. CAR T cells were generated by spinoculation of activated T cells with CAR-encoding lentivirus. In detail, human PBMCs or T cells isolated using a pan T cell isolation kit (130-096-535, Miltenyi Biotec, Inc.) were activated using a human T cell activation/expansion kit (130-091-441, Miltenyi Biotec, Inc.) on day 0. On day 2, T cells were transduced with lentivirus at multiplicities of infection of 3–5 by 1200×g spinoculation for 2 h at 25 °C in media containing 10 μg/mL of polybrene. After spinoculation, the transduced T cells were washed and cultured in medium supplemented with 200 IU/mL of human recombinant IL-2 for 2 weeks. On day 14, CAR-expressing T cells were either used immediately or enriched using anti-FLAG–biotin (130-101-566, Miltenyi Biotec, Inc.) and anti-biotin microbeads (130-091-441, Miltenyi Biotec, Inc.) before use.

**Quantitative PCR**. CAR mRNA expression was determined by quantitative PCR. Total RNA from $1 \times 10^6$ T cells was extracted using an RNeasy plus mini kit (74136, QIAGEN, Hilden, Germany) and reverse-transcribed using the SuperScript III first-strand synthesis system (18080-051, Thermo Fisher Scientific, Inc.). Reverse-transcribed single-stranded DNA was then subjected to quantitative PCR using a FastStart essential DNA green master kit and LightCycler 96 System (06924204001, Roche Molecular Systems, Inc., Basel, Switzerland). CD8TM-BB_Fwd (specific for the junction of the CD8α transmembrane with the 4-1BB signaling domain) and BB-CD3z_Rev (specific for the junction of 4-1BB with the CD3ζ signaling domain) were used to quantify CAR mRNA (Supplementary Table 1). GAPDH_Fwd and GAPDH_Rev (specific for GAPDH mRNA) were used to detect reference gene expression (Supplementary Table 1). CAR mRNA levels relative to GAPDH mRNA levels were calculated and used to compare CAR expression between CAR T cell samples.

**Western blot analysis**. To compare CAR protein levels, we conducted western blot analysis with a CD247-specific antibody (unconjugated; 51-6527GR, BD Biosciences; Supplementary Table 2). In detail, $1 \times 10^7$ T cells were washed three times with ice-cold PBS and lysed with RIPA lysis buffer containing a protease inhibitor cocktail (P3100–001, GenDEPOT, Inc., Barker, TX, USA). The lysates were centrifuged for 10 min at maximum speed at 4 °C and the supernatants were mixed with sample buffer (5×) and boiled for 5 min. Equal amounts of protein were separated on a 12% SDS–PAGE gel and transferred to a polyvinylidene fluoride membrane. The membrane was blocked for 1 h at 25 °C using 5% non-fat milk and incubated in the presence of anti-CD247 antibody overnight at 4 °C with gentle rocking. The membrane was then washed three times with TBS-T buffer and incubated with horseradish peroxidase-conjugated secondary anti-mouse IgG antibody (315–035–045, Jackson ImmunoResearch, Inc., West Grove, PA, USA) and horseradish peroxidase-conjugated β-actin-specific antibody (sc-130656, Santa Cruz Biotechnology, Inc., Dallas, TX, USA) for 1 h at 25 °C. The membrane was washed three times with TBS-T buffer. For signal development, the membrane was developed with a chemiluminescent substrate (NCI4080KR, Thermo Fisher Scientific, Inc.) and exposed to X-ray film. The protein level of CAR relative to β-actin was quantified with ImageJ v1.50i software (NIH, Bethesda, MD, USA).

**Immunofluorescence imaging**. CAR protein localization was assessed by immuno-fluorescence imaging. T cells were fixed in 4% (w/v) paraformaldehyde in PBS (pH 7.4) for 10 min at 25 °C. Fixed cells were washed and permeabilized with perm-wash buffer (PBS, pH 7.4 containing 0.1% saponin and 1% bovine serum albumin) for 20 min at 25 °C and blocked with human Fc Block (564219, BD Biosciences) for 20 min at 25 °C. After washing with perm-wash buffer, the cells were stained with Alexa488-conjugated anti-FLAG-tag antibody (5407, Cell Signaling Technology, Inc., Danvers, MA, USA; Supplementary Table 2) in perm-wash buffer for 30 min at 25 °C. The cells were washed in perm-wash buffer and mounted on glass slides using Vectashield mounting medium containing DAPI (H-1200, Vector Laboratories, Inc., Burlingame, CA, USA) and images were acquired using a Zeiss LSM 780 laser scanning confocal microscope (Carl Zeiss SAS, Oberkochen, Germany).

**Cytotoxicity assays**. Cytotoxic killing of EBV LCLs by T cells was quantified using the CytoTox-Glo cytotoxicity assay kit (G9291, Promega, Madison, WI, USA). In detail, $5 \times 10^4$ EBV LCLs were seeded in 96-well black plates with transparent flat bottoms (3904, Corning, Inc., Corning, NY, USA). T cells were then added to the wells at T cell:EBV LCL ratios of 1:27, 1:9, 1:3, 1:1, or 3:1 and incubated for 4 h at 37 °C. Control wells containing EBV LCLs alone were incubated under the same conditions. After incubation, luminogenic AAF-Glo Substrate was added to each well, and luminescence was measured with a TECAN infinite PRO 200 (Tecan Group, Ltd., Männedorf, Switzerland). Wells containing either EBV LCLs alone or digitonin-treated EBV LCLs were used as controls to detect background and maximum cytotoxicity signals, respectively. Cytotoxicity-induced killing efficacy was determined using the following equation: (cytotoxicity signal in sample well – background cytotoxicity signal)/maximum cytotoxicity signal.

**In vitro on-target killing assay**. To evaluate the target-specific killing efficacy of CAR T cells, a flow cytometry-based killing assay was designed. In detail, PBMCs and EBV LCLs were labeled with a CellTrace violet cell proliferation kit (C34557, Thermo Fisher Scientific, Inc.) and CellTrace carboxyfluorescein succinimidyl ester cell proliferation kit (C34554, Thermo Fisher Scientific, Inc.), respectively. Labeled PBMCs and EBV LCLs were co-cultured with T cells at a T cell:EBV LCL:PBMC ratio of 6:1:1 for 4 h. For co-culture, $1.2 \times 10^6$ cells were incubated in the wells of 48-well plates in 1 mL of medium. Control wells contained labeled EBV LCLs and PBMCs only to measure the decrease in target cells in the absence of T cells. After incubation, 20 μL of Flow-Count fluorospheres (7547053, Beckman Coulter, Inc.) were added to each well for quantitative flow cytometric analysis. The cell–bead mixtures were then transferred into $12 \times 75$-mm polystyrene tubes and stained with the fixable viability dye eFluor 780 (65-0865, Thermo Fisher Scientific, Inc.), and with antibodies specific for HLA-DR, CD14, and CD20. The samples were then fixed with 1% paraformaldehyde and analyzed on a FACSVerse flow cytometer (BD Biosciences). For quantitative population analysis, a fixed number of quantitative beads was acquired from all samples. The killing efficacies of T cells against carboxyfluorescein succinimidyl ester-labeled EBV LCLs and violet-labeled CD20 B cells were calculated using the following equation: EBV LCL-killing efficacy = (live EBV LCLs in control well – live EBV LCLs in sample well)/ live EBV LCLs in control wells; B cell killing efficacy = (live B cells in control well – live B cells in sample well)/live B cells in control wells.

**Cytotoxicity inhibition assay**. The cytotoxicity inhibition assay was performed as in the in vitro on-target killing assay with some modifications. Briefly, EBV LCLs were labeled using a CellTrace violet cell proliferation kit (C34557, Thermo Fisher Scientific, Inc.) and co-cultured with each type of T cell at a T cell:EBV LCL ratio of 5:1 for 4 h in the presence of anti-CD178 (FasL) antibody (FasL blocker; uncon-jugated; 10 μg/mL; 556371, BD Biosciences; Supplementary Table 2), anti-CD253 (TRAIL) antibody (TRAIL blocker; unconjugated; 10 μg/mL; 550912, BD Bios-ciences; Supplementary Table 2), concanamycin A (CMA; perforin-1 blocker; 1 μg/mL; C9705-25UG, Sigma-Aldrich, St. Louis, MO, USA), or recombinant human Bcl-2 Protein (granzyme B blocker; 1 μg/mL; 827-BC, R&D Systems, Minneapolis, MN, USA). Samples of $1.2 \times 10^6$ cells were co-cultured in 48-well plates with 0.5 mL media. A T cell–EBV LCL mixture, containing 10 μg/mL of isotype mouse IgGs and 1 μg/mL of dimethyl sulfoxide, was used as a non-inhibited control. Labeled EBV LCLs alone were used as background controls. After incubation, 20 μL of Flow-Count fluorospheres (7547053, Beckman Coulter, Inc.) were added directly to each well for quantitative flow cytometric analysis. The cell–bead mixtures were then transferred to $12 \times 75$-mm polystyrene tubes and stained with fixable viability dye eFluor780 (65-0865, Thermo Fisher Scientific, Inc.), and then fixed with 1% paraformaldehyde and analyzed on a FACSVerse flow cytometer (BD Biosciences). For quantitative analysis, a fixed number of quantitative beads were acquired from all samples. The efficiency of inhibited EBV LCL killing was determined using the following equation: (EBV LCLs in reagent-containing sample – EBV LCLs in non-inhibited controls)/(EBV LCLs in background control – EBV LCLs in non-inhibited controls).

**Counting surface molecules**. Surface molecules were quantified using a Quantum Simply Cellular anti-Mouse IgG kit (814, Bangs Laboratories, Inc., Fishers, IN, USA). APC-conjugated FLAG-specific antibodies, PE-conjugated CD19-specific

antibodies, and PE-Cy5-conjugated HLA-DR-specific antibodies were used to quantify CAR, CD19, and HLA-DR, respectively. Flow cytometric analysis was performed using a FACSVerse flow cytometer (BD Biosciences).

**Measurement of granule transfer rates**. Granule transfer rates following contact between T cells and B cells (or EBV LCLs) were measured by flow cytometry. First, T cells were labeled with a CellTrace violet cell proliferation kit (C34557, Thermo Fisher Scientific, Inc.). EBV LCLs or B cells from the PBMCs of healthy donors isolated using a B cell isolation kit II (130–091–151, Miltenyi Biotec, Inc.) were used as target cells. Samples of $4.5 \times 10^5$ T cells and target cells in a T cell:target cell ratio of 2:1 were incubated for 10, 30, or 90 min in 96-well flat bottom plates. After incubation, the cells were fixed and permeabilized with a Cytofix/Cytoperm Fixation/Permeabilization kit (554714, BD Biosciences) and transferred granules were stained with anti-granzyme A and anti-granzyme B antibodies and analyzed by FACSVerse flow cytometer (BD Biosciences). The target cells were identified by gating on violet-negative cells (Supplementary Fig. 6a). The granule-transfer rate was calculated from the percentage of granzyme A and/or granzyme B-positive cells among the total target cells.

**Live imaging of apoptotic cells**. The kinetics of EBV LCL apoptosis were mea-sured with a JuLI Stage real-time cell history recorder (NanoEnTek, Inc., Gyeonggi-do, Korea). Target EBV LCLs were labeled with a CellTrace violet cell proliferation kit (C34557, Thermo Fisher Scientific, Inc.). Samples of $1 \times 10^5$ T cell and EBV LCL at a T cell:EBV LCL ratio of 1:1 were incubated in 96-well flat-bottom plates in the presence of IncuCyte caspase-3/7 reagent to induce apoptosis (4440, Essen BioScience, Ann Arbor, MI, USA). DAPI- and RFP-filtered images were taken every 5 min for 90 min. Three areas of each well were analyzed. Because of the blue fluorescence of violet-labeled EBV LCLs, apoptotic EBV LCLs can be identified by observing magenta-colored cells in merged images (blue fluorescence of violet label combined with red fluorescence of apoptotic cells). The percentage of apoptotic EBV LCLs was determined and converted into a numerical value with ImageJ v1.50i software and JuLI STAT (NanoEnTek, Inc.). The proportion of apoptotic EBV LCLs was calculated from the equation: % apoptotic EBV LCLs = apoptotic EBV LCLs (magenta colored)/total EBV LCLs (blue or magenta colored).

**Animal models**. For all animal experiments, we used immune-deficient 7–10-week-old $C;129S4$-$Rag2^{tm1.1Flv}Il2rg^{tm1.1Flv}/J$ female mice kept under specific-pathogen-free conditions. Mice were sacrificed by carbon dioxide exposure when tumor volume exceeded 2000 mm³ or the total luminescence of luciferin-treated subject exceeded $1 \times 10^{11}$ photons/s. This study was approved by the Institutional Animal Care and Use Committee of the National Cancer Center Research Institute. The National Cancer Center Research Institute is a facility accredited by the Association for Assessment and Accreditation of Laboratory Animal Care Inter-national and abides by the Institute of Laboratory Animal Resources guidelines.

**In vivo efficacy test**. In vivo CAR T cell efficacy was evaluated using a xenograft model. Five days before T cell infusion, mice were intraperitoneally xenografted with $3 \times 10^6$ (100 μL) luciferase-expressing EBV LCL-lucH cells. After 5 days (on day 0), $5 \times 10^6$ T cells (300 μL) were injected intravenously per mouse. Four mice were injected with NT T cells, and five mice were injected with CD19 CAR T and MVR CAR T cells, respectively. The tumor burdens of the xenografted mice were determined on days 0, 7, 14, 21, and 28 by measuring luciferase activity with an IVIS Lumina in vivo imaging system (PerkinElmer, Inc., Waltham, MA, USA).

**In vivo on-target killing assay**. A transient xenograft model was used for assaying in vivo on-target killing. In detail, 1 mg of clodronate liposomes (ClodLip BV, Amsterdam, Netherlands) was injected intravenously into mice 5 days before infusion with T cells. The next day, the mice were X-ray irradiated with a dose of 2 Gy using X-RAD 320 (Precision X-Ray, Inc., North Branford, CT, USA), and intravenously grafted with $3 \times 10^5$ (300 μL) $DR^{weak}$ B cells from $DR^{weak}$ PBMCs obtained with a B cell isolation kit II (130-091-151, Miltenyi Biotec, Inc.). Three days before T cell infusion, $6.5 \times 10^5$ (200 μL) of luciferase-expressing EBV LCL-lucH cells were injected intraperitoneally into the mice. After 3 days (on day 0) $1 \times 10^7$ T cells (500 μL) were injected intravenously per mouse. Four mice were injected with NT T and MVR CAR T cells, respectively, and five mice were injected with CD19 CAR T cells. All xenografted mice were analyzed for tumor burden on days −1, 7, and 14 by measuring luciferase activity with the IVIS Lumina in vivo imaging system. The persistence of B cells and blood IFN-γ levels were measured in blood samples collected by retro-orbital bleeding on day −1, 2, and 7. To quantify the remaining B cells in blood samples, CD3-, CD20-, CD45-, and HLA-DR-specific antibodies were added directly to 75 μL of EDTA-treated peripheral blood. After staining, red blood cell lysis buffer was added and the samples were transferred into $12 \times 75$-mm polystyrene tubes. Flow-Count fluorospheres (7547053, Beckman Coulter, Inc.) were added to each well for quantitative flow cytometric analysis. The cell–bead mixtures were then washed twice and fixed with 1% paraformaldehyde, and analyzed by FACSVerse flow cytometry. For quantitative population analysis, a fixed number of quantitative beads were acquired from all samples. IFN-γ levels in plasma collected from centrifuged blood samples were quantified with a BD

Cytometric Bead Array human Th1/Th2/Th17 cytokine kit (560484, BD Biosciences).

**Statistical analyses**. We used statistical tests appropriate for the data based on similar studies in the field. Unpaired two-tailed *t*-tests were used to evaluate differences unless otherwise specified; $p < 0.05$ was considered statistically significant and significance is designated with asterisks (ns, not significant; *$p < 0.05$; **$p < 0.01$; ***$p < 0.001$). Prism v5.01 (GraphPad Software, Inc., La Jolla, CA, USA) was used to generate all graphs and for all statistical analyses.

**Study design**. Sample sizes were determined based on similar studies without the use of statistical methods. No samples or animals were excluded from analysis. No blinding or randomization was used during the experiments.

**Data availability**. The data that support the findings of this study are available from the corresponding author upon reasonable request.

Published online: xx xxx 2018

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

## Acknowledgements

We thank Dr. Carl June for providing pELPS-19BBz. This work was supported by grants from the National Cancer Center of Korea (NCC-1610320), National Research Foundation of Korea (NRF-2005-0093837), and Korean Ministry of Trade, Industry and Energy (GLOBAL R&D PROJECT, N0000901). We also thank the Flow Cytometry Core Facility, Fluorescence Imaging Core Facility, Molecular Imaging & Therapy Branch, and Animal Sciences Branch of the National Cancer Center Research Institute for technical support.

## Author contributions

C.H. conceived the project, designed, and conducted most of the experiments, and wrote the manuscript. S.-J.S. and S.L. contributed to the generation of lentivirus and CAR T cells. R.S. and K.H.K. performed immunofluorescence microscopy and interpreted the data. S.H. contributed to western blot analyses. Y.H.K. generated EBV. S.-H.K. and B.K. C. contributed to the design of the animal experiments and their analysis. S.-H.K., Y.I.K., S.H.P., D.G.L., H.S.O., Y.H.K., and B.K.C. assisted in the animal experiments. B.S.K. supervised the project, contributed to the design of the experiments and their analysis, and wrote the manuscript.

## Additional information

**Competing interests:** B.S.K. is the founder and Chief Executive Officer of Eutilex Co., Ltd. The remaining authors declare no competing financial interests.

