## [Peer Review File · Nature Communications]

Reviewers' comments:

Reviewer #1 (T activation, anti-tumor immunity)(Remarks to the Author):

The study by Han et al have examined the functional consequences of T cells transduced with high avidity or low avidity CAR constructs. They have convincingly demonstrated that T cells expressing high avidity CARs undergo fratricide while lower avidity CARs can be expressed, can control tumor growth in an adoptive transfer/mouse model. The model described in this paper, is elegant and experiments are well performed. These findings are relevant for designing CAR constructs, and predictions for efficacy of therapy.

Minor comments:

1). Some figures are small and hard to read (figure 5 fg).

Reviewer #2 (Cancer therapy, CAR-T)(Remarks to the Author):

The proposed article describes the use of a developed antibody (MVR) specific to HLA-DR for its potential use in CAR therapy. The researchers demonstrate that donors with HLA-DR that bind with weak affinity to the MVR antibody (DR-weak) were able to produce MVR CAR T-cells; while donors with intermediate or strong binding HLA-DR quickly induced fratricide. Even in DR-weak donors, over an extended culture period, T cells downregulated CAR expression. These "autotuned" T cells only elicited a response against EBV transformed B cells, as opposed to normal B cells, due to an associated increase in HLA-DR expression.

Simply, the proposed article describes the influence of protein expression on the interaction between a CAR and its cognate antigen. Specifically, it describes how CAR T cells that target HLA-DR, which is endogenously expressed on activated T cells, down-regulate CAR expression until desensitized to the presence of HLA-DR. This phenomenon in turn restricts CAR T cell activity to high HLA-DR expressing target cells. Structurally, the experiments performed in the proposed article follow a logical progression that seeks to define the CAR:antigen interaction. The figures eloquently demonstrate the specific activity of DRweak MVR CAR T cells against HLA-DR upregulated, DRweak EBV transformed B cells.

The interplay between CAR and antigen expression levels as it relates to therapeutic activity has been described in multiple articles. For example, Federov et al (PMID 24337479) describe the stoichiometric relationship of cytotoxicity, cytokine production, and CAR expression. They further describe the influence of antigen expression levels on CAR activity. Similarly Chang et al (PMID 25990251) define functional differences between CD19CAR high and low T cells. Again, antigen expression levels directly affected CAR activity in PMID 28784559 and 27197068. While the information produced in the proposed article may be of interest to those in the direct CAR community, it is analogous to prior publications and may not be broadly applicable to a wider field.

While the authors efficiently summarized the interaction between the MVR CAR and its target HLA-DR, this combination is limited in scope, as the authors themselves describe, to patients with DR weak HLA-DR alleles. The success and specificity of the described approach is also contingent upon the stable up-regulation of HLA-DR in malignant B-cells during therapy. From the described experiments it is difficult to decipher a threshold where normal HLA-DR expressing B cells are protected from on-target off-tumor side effects. These studies could be bolstered by a more in depth quantitation of CAR levels and HLA-DR levels on all target cells used in the experiments. This could easily be performed via the use of quantitation beads and flow cytometry. A comparative analysis of primary and malignant B cells from patient samples would further ready this article for publication and be more applicable to a larger audience.

Reviewer #3 (CAR-T, TCR-T, cancer therapy)(Remarks to the Author):

The manuscript by Han et al. describes a series of interesting observations about the functional impact of affinity of CAR constructs for target antigens. The authors utilize polymorphic recognition of HLA-DR by HLA-DR-specific CAR to generate CAR engineered cells that recognize target cells strongly, moderately, and weakly, and characterize the phenotype and function of these cells. The experimental system is clever, and the results are interesting, but ultimately predictable: due to fratricide, only engineered cells that weakly recognize targets are able to be functionally characterized in any detail. Ultimately this work highlights the relevance of affinity and avidity on effective target recognition, but fails to inform on any novel mechanisms that can be exploited to develop more effective CAR therapies.

Response to the Reviewers' comments

We are very grateful to the reviewers for their valuable suggestions for improving our manuscript. Based on these suggestions we have conducted additional experiments and analyses, and have revised the manuscript. We believe that the manuscript has been greatly improved as a result, and that the novelty of the current findings has been made more apparent. We have responded to the best of our ability to the concerns raised by the reviewers, and provide below a point-by-point response to these concerns.

Reviewers' comments:

Reviewer #1 (T activation, anti-tumor immunity)(Remarks to the Author):

The study by Han et al have examined the functional consequences of T cells transduced with high avidity or low avidity CAR constructs. They have convincingly demonstrated that T cells expressing high avidity CARs undergo fratricide while lower avidity CARs can be expressed, can control tumor growth in an adoptive transfer/mouse model. The model described in this paper, is elegant and experiments are well performed. These findings are relevant for designing CAR constructs, and predictions for efficacy of therapy.

Minor comments:

1). Some figures are small and hard to read (figure 5 fg).

Author response: We appreciate the positive comments. In response to the reviewer's suggestion, we have increased the size of figure 5f and g (page 36).

Reviewer #2 (Cancer therapy, CAR-T)(Remarks to the Author):

The proposed article describes the use of a developed antibody (MVR) specific to HLA-DR for its potential use in CAR therapy. The researchers demonstrate that donors with HLA-DR that bind with weak affinity to the MVR antibody (DR-weak) were able to produce MVR CAR T-cells; while donors with intermediate or strong binding HLA-DR quickly induced fratricide. Even in DR-weak donors, over an extended culture period, T cells downregulated CAR expression. These "autotuned" T cells only elicited a response against EBV transformed B cells, as opposed to normal B cells, due to an associated increase in HLA-DR expression.

Simply, the proposed article describes the influence of protein expression on the interaction between a CAR and its cognate antigen. Specifically, it describes how CAR T cells that target HLA-DR, which is endogenously expressed on activated T cells, down-regulate CAR expression until desensitized to the presence of HLA-DR. This phenomenon in turn restricts CAR T cell activity to high HLA-DR expressing target cells. Structurally, the experiments performed in the proposed article follow a logical progression that seeks to define the CAR:antigen interaction. The figures eloquently demonstrate the specific activity of DRweak MVR CAR T cells against HLA-DR upregulated, DRweak EBV transformed B cells.

The interplay between CAR and antigen expression levels as it relates to therapeutic activity has been described

in multiple articles. For example, Federov et al (PMID 24337479) describe the stoichiometric relationship of cytotoxicity, cytokine production, and CAR expression. They further describe the influence of antigen expression levels on CAR activity. Similarly Chang et al (PMID 25990251) define functional differences between CD19CAR high and low T cells. Again, antigen expression levels directly affected CAR activity in PMID 28784559 and 27197068. While the information produced in the proposed article may be of interest to those in the direct CAR community, it is analogous to prior publications and may not be broadly applicable to a wider field.

Author response: We appreciate the careful review of our manuscript and the relevant information and references. As the reviewer mentions, the relation between T cell activity and the expression levels of CAR and antigen has been examined previously. Fedorov et al.¹ and Chang et al.² demonstrated that CAR T cells sorted for higher CAR expression had stronger effector functions than those sorted for low CAR expression. Furthermore, Fedorov et al.¹ and Han et al.³ reported that antigen expression level and CAR T cell activity were correlated. Other studies also showed that the correlation was more significant when CAR T cells and blocking antibody were used together⁴ or the CARs had low affinity^{5,6}.

While these papers describe observations similar to those in the current study, our findings differ in two important respects. First, we describe an auto-desensitization phenomenon (“autotuning”) that modulates CAR T cell reactivity: the MVR CAR T cells used throughout the current study recognized HLA-DR on their cell surfaces and induced sustained downregulation of the CAR, and this in turn led to improved target cell discrimination based on the level of expression of HLA-DR. Unlike the studies of Fedorov et al.¹ and Chang et al.², where CAR^{low} T cells were sorted based on CAR expression level, we investigated this CAR downregulation phenomenon. Further studies elucidating the conditions in which CAR expression is optimally downregulated will help to generate safer CAR T cells. Second, our findings emphasize the advantage of combining a low affinity CAR with low expression of the CAR, which dramatically improves target cell selectivity. Fedorov et al.¹ and Chang et al.² reported that low CAR expression significantly decreased CAR T cell activity towards the target cells. Furthermore, Liu et al.⁵ and Caruso et al.⁶ showed that low CAR affinity altered the magnitude of CAR T cell activity. In our case, the autotuned MVR CAR T cells had both low affinity and low CAR expression, and exhibited far better target cell selectivity based on antigen level. In particular, the MVR CAR T cells efficiently discriminated primary B cells from EBV-transformed B cells in *in vitro* and *in vivo* (**Fig. 3e,4f,5f**). The low affinity and low CAR expression were both important since DR^{weak} B cells were killed by untuned MVR CAR T cells (low affinity only; **Fig. 4e**) and DR^{str} B cells were killed by autotuned MVR CAR T cells (low CAR expression only; **Fig. 4e**). We believe that these two points confer substantial novelty on the current study. We regret that these points were insufficiently emphasized in the original manuscript. We now underscore them in the discussion and add the suggested references (page 11, line 21 to page 12, line 14).

While the authors efficiently summarized the interaction between the MVR CAR and its target HLA-DR, this combination is limited in scope, as the authors themselves describe, to patients with DR weak HLA-DR alleles. The success and specificity of the described approach is also contingent upon the stable up-regulation of HLA-

DR in malignant B-cells during therapy. From the described experiments it is difficult to decipher a threshold where normal HLA-DR expressing B cells are protected from on-target off-tumor side effects. These studies could be bolstered by a more in depth quantitation of CAR levels and HLA-DR levels on all target cells used in the experiments. This could easily be performed via the use of quantitation beads and flow cytometry.

Author response: As the reviewer points out, identifying a threshold value that minimizes on-target off-tumor side effects is of great interest. Therefore, we have carefully measured CAR/HLA-DR levels on the effector and target cells. The median count of CAR molecules expressed on MVR CAR T cells was $698,123 \pm 7,458$ (mean \pm s.e.m., $n = 4$) in the untuned condition, and it decreased to $124,854 \pm 2,531$ (mean \pm s.e.m., $n = 4$) in the autotuned cells (**Fig. 4d**). The median count of HLA-DR molecules on normal B cells was $42,590 \pm 2,458$ (mean \pm s.e.m., $n = 4$), and this increased to $82,945 \pm 6,395$ (mean \pm s.e.m., $n = 4$) and $78,513 \pm 8,963$ (mean \pm s.e.m., $n = 4$) per cell in activated B cells and EBV-transformed B cells, respectively (**Fig. 4b,c**). Thus autotuning decreased CAR expression ~ 6 -fold, and activation and EBV-transformation increased HLA-DR expression ~ 2 -fold.

Recently, Taylor et al. investigated the quantitative requirements for T-cell signaling in a DNA-CAR system⁷. The system used a CAR construct consisting of a DNA binding motif and TCR ζ , so that antigen quantity and CAR–antigen affinity could be tightly regulated. Importantly, they performed stochastic simulations based on the observed results and found that, in some conditions, ~ 3 -fold changes in ligand density or affinity resulted in a sharp transition to receptor clustering leading to T cell signaling and function. Consistent with that study, we observed that ~ 2 -fold changes in HLA-DR expression level in the target cells led to cytotoxic killing by the CAR T cells (**Fig. 4c,e**). Furthermore, we observed that ~ 6 -fold changes in CAR expression level enhanced the MVR CAR–HLA-DR interaction, and also resulted in target cell killing (**Fig. 4d,e**). The acute sensitivity to the levels of HLA-DR and MVR CAR depends on low affinity, since DR^{str} HLA-DR-expressing cells were eliminated by MVR CAR T cells regardless of the expression level of HLA-DR and CAR (**Fig. 3e,4e**). Collectively, the current study highlights the requirements for minimizing unwanted killing by CAR T cells under physiological condition: weak affinity, use of an antigen whose expression is > 2 -fold higher in the target malignant cells, and reduction of the CAR expression level by some appropriate process such as autotuning.

We have added these points and the relevant experimental results to the result/discussion sections and figures (page 8, line 14 to page 9, line 1; page 12, line 25 to page 13, line 13; **Fig. 4c,d**). We believe that these changes clearly highlight the novel findings of the current study and provide insight into the relation between avidity and the function of CAR T cells. We thank the reviewer for the suggestion, which as substantially improved the quality of the revised manuscript.

A comparative analysis of primary and malignant B cells from patient samples would further ready this article for publication and be more applicable to a larger audience.

Author response: We appreciate the reviewer’s comment and valuable suggestion. In the current study, we sought to establish the effects of autotuning and CAR affinity on target cell selectivity. From a basic research perspective, our study is relevant to developing strategies that minimize unwanted on-target off-tumor side effects by modulating CAR expression levels and selecting appropriate target antigens. We agree that the

therapeutic potential of HLA-DR-targeted MVR CAR T cells in B cell malignancies was not adequately covered from an application-oriented perspective. Therefore to investigate the applicability of MVR CAR T cells in B cell malignancies, we quantitated HLA-DR expression levels in various malignant cell lines, as the reviewer suggested (**Supplementary Fig. 8**). Interestingly, 3 out of 7 cell lines exhibited similar or higher levels of surface HLA-DR expression than the EBV-transformed B cells used in the current study (**Fig. 4b**). This underlines the therapeutic potential of MVR CAR T cells in treating HLA-DR-upregulated B cell malignancies with reduced on-target off-tumor side effects, as shown in the EBV-transformed lymphoma model in the current study.

We have add these data to the figure and the manuscript (**Supplementary Fig. 8**; page 12, line 15-24). We believe that the revised manuscript now provides valuable information not only to individuals interested in CAR basic research but also to those working in clinical areas.

Reviewer #3 (CAR-T, TCR-T, cancer therapy) (Remarks to the Author):

The manuscript by Han et al. describes a series of interesting observations about the functional impact of affinity of CAR constructs for target antigens. The authors utilize polymorphic recognition of HLA-DR by HLA-DR-specific CAR to generate CAR engineered cells that recognize target cells strongly, moderately, and weakly, and characterize the phenotype and function of these cells. The experimental system is clever, and the results are interesting, but ultimately predictable: due to fratricide, only engineered cells that weakly recognize targets are able to be functionally characterized in any detail. Ultimately this work highlights the relevance of affinity and avidity on effective target recognition, but fails to inform on any novel mechanisms that can be exploited to develop more effective CAR therapies.

Author response: As the reviewer says, the current study primarily sought to establish the role of CAR affinity in CAR T cell function. To reduce the bias of tonic signaling by different CAR frames⁸, we used one fixed MVR CAR frame and a series of antigens with different binding affinities. However, since T cell-derived HLA-DR expression led to serious fratricide in the strong/intermediate affinity cases, only DR^{weak} MVR CAR T cells were used in the study, and we assessed the impact of affinity on effector function indirectly using two combinations of effector/target cells (i.e. DR^{weak} MVR CAR T cells + DR^{weak} EBV LCLs or DR^{weak} MVR CAR T cells + DR^{str} EBV LCLs; **Fig. 3a,e**).

Although the auto-HLA-DR recognition by MVR CAR T cells and the subsequent fratricide narrows the focus on the relation between affinity and CAR T cell function, it has revealed novel findings regarding weak interaction-induced CAR downregulation and its subsequent functional impact. Previous studies have described the impact of CAR affinity on target recognition and the functional changes in CAR T cells^{5,6}. Along with confirming this, our study further shows that the combination of weak affinity and CAR downregulation improves target cell selectivity. Specifically, MVR CAR T cells efficiently discriminated primary B cells from EBV-transformed B cells *in vitro* and *in vivo* (**Fig. 3e,5f**). Of note, the low affinity and low CAR expression were both important since DR^{weak} B cells were killed by untuned MVR CAR T cells (low affinity only; **Fig. 4e**) and DR^{str} B cells were killed by autotuned MVR CAR T cells (low CAR expression only; **Fig. 4e**). Most importantly, our findings focus attention on “autotuning” as a novel strategy for downregulating CAR

expression. In the autotuning process, MVR CAR T cells recognize HLA-DR on their surfaces and gradually reduce their reactivity by mild CAR downregulation. Since this autonomous CAR downregulation reveals the self-limiting behavior of the CAR T cells that leads to reduced on-target off-tumor side effects, we believe that our study provides important information for workers in the field of CAR T cell development.

We regret that these issues were insufficiently emphasized in the original manuscript, and we now consider them in the revised discussion (page 11, line 21 to page 12, line 14).

References

- 1 Fedorov, V. D., Themeli, M. & Sadelain, M. PD-1- and CTLA-4-based inhibitory chimeric antigen receptors (iCARs) divert off-target immunotherapy responses. *Sci. Transl. Med.* **5**, 215ra172, doi:10.1126/scitranslmed.3006597 (2013).
- 2 Chang, Z. L., Silver, P. A. & Chen, Y. Y. Identification and selective expansion of functionally superior T cells expressing chimeric antigen receptors. *J. Transl. Med.* **13**, 161, doi:10.1186/s12967-015-0519-8 (2015).
- 3 Han, X. *et al.* Adnectin-Based Design of Chimeric Antigen Receptor for T Cell Engineering. *Mol. Ther.* **25**, 2466-2476, doi:10.1016/j.ymthe.2017.07.009 (2017).
- 4 Rufener, G. A. *et al.* Preserved Activity of CD20-Specific Chimeric Antigen Receptor-Expressing T Cells in the Presence of Rituximab. *Cancer Immunol. Res.* **4**, 509-519, doi:10.1158/2326-6066.CIR-15-0276 (2016).
- 5 Liu, X. *et al.* Affinity-Tuned ErbB2 or EGFR Chimeric Antigen Receptor T Cells Exhibit an Increased Therapeutic Index against Tumors in Mice. *Cancer Res.* **75**, 3596-3607, doi:10.1158/0008-5472.CAN-15-0159 (2015).
- 6 Caruso, H. G. *et al.* Tuning Sensitivity of CAR to EGFR Density Limits Recognition of Normal Tissue While Maintaining Potent Antitumor Activity. *Cancer Res.* **75**, 3505-3518, doi:10.1158/0008-5472.CAN-15-0139 (2015).
- 7 Taylor, M. J., Husain, K., Gartner, Z. J., Mayor, S. & Vale, R. D. A DNA-Based T Cell Receptor Reveals a Role for Receptor Clustering in Ligand Discrimination. *Cell* **169**, 108-119 e120, doi:10.1016/j.cell.2017.03.006 (2017).
- 8 Long, A. H. *et al.* 4-1BB costimulation ameliorates T cell exhaustion induced by tonic signaling of chimeric antigen receptors. *Nat. Med.* **21**, 581-590, doi:10.1038/nm.3838 (2015).

REVIEWERS' COMMENTS:

Reviewer #2 (Remarks to the Author):

The author's rebuttal provides helpful clarifications and some additional supplemental data. However, as pointed out in the review, the data falls short of a definitive body of work that merits publication in *Mature Communications*.

Reviewer #3 (Remarks to the Author):

The revised manuscript by Han et al. fails to convince this reviewer that the reported observations are anything more substantial than phenomenology that is unique to targeting a surface marker expressed on T cells with activation. The observations could simply reflect the consequence of a selection process for T cells to escape fratricide, without any more profound biology. It is extremely unlikely that the reported observations bear any semblance to thymic sensitivity selection. It is also unclear how this observation has any generalizable clinical relevance or translatability, since the empiric nature of the fratricide cannot be assumed to translate to other targets

Response to the Reviewers' comments

Reviewers' comments:

Reviewer #2 (Remarks to the Author):

The author's rebuttal provides helpful clarifications and some additional supplemental data. However, as pointed out in the review, the data falls short of a definitive body of work that merits publication in Mature Communications.

Author response: Primarily, the current study investigated the relationship between the expression level of antigen, CAR, and affinity, which affects the resulting functional differences in CAR T cells. As described in our previous rebuttal, our study emphasizes the enhanced antigen-sensing potential of the combination of weak affinity with adequately adjusted CAR expression level, which distinguishes our study from previous studies reporting the individual effects of affinity or CAR expression levels. Most importantly, our study investigated the 'T cell-like' properties of CAR T cells and the resulting functional consequences. MVR CAR T cells were simply generated by transduction of T cells with HLA-DR-targeting CAR, and subsequent events were induced by T cell-intrinsic function. Activated T cells showed upregulated HLA-DR on the surface, which led to CAR downregulation and fratricide by HLA-DR–MVR CAR interactions. This eventually resulted in the desensitization of MVR CAR T cells and enhanced antigen-sensing ability. Considering that most CAR T cell studies have focused on excavating novel targets and CAR constructs, our study provides a new perspective showing that CAR T cells can be adequately desensitized using T cell-intrinsic machinery through *ex vivo* conditioning to reduce harmful side effects that frequently occur during CAR T cell therapy.

Reviewer #3 (Remarks to the Author):

The revised manuscript by Han et al. fails to convince this reviewer that the reported observations are anything more substantial than phenomenology that is unique to targeting a surface marker expressed on T cells with activation. The observations could simply reflect the consequence of a selection process for T cells to escape fratricide, without any more profound biology. It is extremely unlikely that the reported observations bear any semblance to thymic sensitivity selection. It is also unclear how this observation has any generalizable clinical relevance or translatability, since the empiric nature of the fratricide cannot be assumed to translate to other targets.

Author response: As mentioned by the reviewer, our study describes the generation of HLA-DR-targeted MVR CAR T cells (e.g. fratricide) as well as the functional consequences. The fratricidal selection described in this study is interesting from a basic research-oriented perspective, but difficult to translate into a generalized strategy for CAR T cell development. However, in addition to fratricide, our study addresses the 'tuning' potential of CAR T cells. Because MVR CAR continuously recognized HLA-DR on the T cell surface, MVR CAR expression was downregulated by the HLA-DR–MVR CAR interaction, which was repeatedly observed

throughout the study. Notably, downregulation was not transient, but was sustained for over 4 weeks (**Supplementary Fig. 8**). Furthermore, CAR downregulation directly affected B cell/EBV LCL killing by CAR T cells (**Fig. 3f**). This indicates that the CAR expression level was adjusted by the initial interaction between CAR and the target antigen, resulting in the tuning of CAR T cell reactivity at the macro level. Collectively, the current study revealed that the potential of long-term desensitization of CAR T cells by initial *ex vivo* conditioning, providing a foundation for the study and development of CAR T cells in the future.